# PRMT2 promotes HIV-1 latency by preventing nucleolar exit and phase separation of Tat into the Super Elongation Complex

Jiaxing Jin[1,8], Hui Bai[1,8], Han Yan[1], Ting Deng[2], Tianyu Li[3], Ruijing Xiao[3], Lina Fan[4], Xue Bai[5], Hanhan Ning[1], Zhe Liu[5], Kai Zhang [5], Xudong Wu [5], Kaiwei Liang [3], Ping Ma[4] ✉, Xin Gao [6,7] ✉ & Deqing Hu [1] ✉

The HIV-1 Tat protein hijacks the Super Elongation Complex (SEC) to stimulate viral transcription and replication. However, the mechanisms underlying Tat activation and inactivation, which mediate HIV-1 productive and latent infection, respectively, remain incompletely understood. Here, through a targeted complementary DNA (cDNA) expression screening, we identify PRMT2 as a key suppressor of Tat activation, thus contributing to proviral latency in multiple cell line latency models and in HIV-1-infected patient CD4[+] T cells. Our data reveal that the transcriptional activity of Tat is oppositely regulated by NPM1-mediated nucleolar retention and AFF4-induced phase separation in the nucleoplasm. PRMT2 preferentially methylates Tat arginine 52 (R52) to reinforce its nucleolar sequestration while simultaneously counteracting its incorporation into the SEC droplets, thereby leading to its functional inactivation to promote proviral latency. Thus, our studies unveil a central and unappreciated role for Tat methylation by PRMT2 in connecting its subnuclear distribution, liquid droplet formation, and transactivating function, which could be therapeutically targeted to eradicate latent viral reservoirs.

Although combination antiretroviral therapy (ART) potently suppresses HIV-1 replication and allows infected individuals to lead normal lives, the persistence of chronic infections caused by latent reservoirs in patients renders HIV-1 infection incurable. These latent reservoirs are primarily comprised of memory CD4[+] T cells that harbor genetically intact, transcriptionally inactive but replication-competent HIV-1 provirus that can rapidly rebound into reproductive replication upon ART interruption, leading to a requirement of lifelong ART. One of the strategies proposed to achieve a functional cure for HIV-1 infection is the "shock and kill" approach, which aims to eliminate latent reservoirs by reactivating the replication of latent provirus with latency-reversing agents (LRAs) to trigger viral cytopathic effect or immune clearance.

[1]National Clinical Research Center for Cancer, Tianjin's Clinical Research Center for Cancer, Key Laboratory of Cancer Prevention and Therapy, The Province and Ministry Co-sponsored Collaborative Innovation Center for Medical Epigenetics, State Key Laboratory of Experimental Hematology, Key Laboratory of Immune Microenvironment and Disease of Ministry of Education, Department of Cell Biology, School of Basic Medicine, Tianjin Medical University Cancer Institute and Hospital, Tianjin Medical University, 300070 Tianjin, China. [2]Key Laboratory of Breast Cancer Prevention and Therapy of Ministry of Education, Key Laboratory of Cancer Prevention and Therapy, Tianjin Medical University Cancer Institute and Hospital, 300060 Tianjin, China. [3]Department of Pathophysiology, School of Basic Medical Sciences,  Wuhan University, 430071 Wuhan, China. [4]Department of Infectious Diseases, Tianjin Second People's Hospital, Nankai University, 300192 Tianjin, China. [5]The Province and Ministry Co-sponsored Collaborative Innovation Center for Medical Epigenetics, School of Basic Medical Sciences, Tianjin Medical University, 300070 Tianjin, China. [6]State Key Laboratory of Experimental Hematology, National Clinical Research Center for Blood Diseases, Haihe Laboratory of Cell Ecosystem, Institute of Hematology & Blood Diseases Hospital, Chinese Academy of Medical Sciences & Peking Union Medical College, 300020 Tianjin, China. [7]Tianjin Institutes of Health Science, 301600 Tianjin, China. [8]These authors contributed equally: Jiaxing Jin, Hui Bai. ✉e-mail: docmaping@outlook.com; gaoxin1@ihcams.ac.cn; hudq@tmu.edu.cn

Although a few LRAs with distinct mechanisms of action, such as protein kinase C (PKC) agonists, histone deacetylase inhibitors, and the bromodomain-containing protein 4 (BRD4) inhibitors, have been shown to reactivate latent provirus in cell lines and primary CD4+ T cell models of HIV-1 latency, clinical trials using these agents have failed to significantly reduce the size of latent reservoirs in the peripheral blood of infected patients, suggesting insufficient potency of these LRAs in latency reversal and warranting further investigations into the molecular mechanisms for the establishment, maintenance and reactivation of integrated provirus[1].

The mechanisms underlying HIV-1 transcription have been intensively studied and involve all transcriptional steps that are coordinately regulated by cellular and viral factors. In particular, promoter-proximal pausing of RNA polymerase II (hereafter Pol II), which stalls after synthesis of 30–50 nucleotide viral transcript, is a major rate-limiting step in HIV-1 gene transcription[2]. The escape of paused Pol II into productive elongation requires the collaborative efforts of the viral transactivator Tat and cellular elongation factors, such as p-TEFb, AFF4, ELL2, and ENL or AF9, which assemble into a large and stable Super Elongation Complex (SEC)[3,4]. Through directly binding to the transactivation response element (TAR) located in the 5′ end of the nascent viral transcript, Tat recruits the SEC to the viral promoter to enable rapid release of paused Pol II into processive elongation by phosphorylating the consensus heptad repeat of YSPTSPS on the carboxyl terminal domain (CTD) of Pol II and additional cellular factors, including negative elongation factor (NELF), DRB sensitivity-inducing factor (DSIF), and others[3,5]. Nevertheless, the molecular mechanism behind how Tat recruitment of SEC leads to the highly efficient elongation of Pol II on the HIV-1 gene remains unclear. Recent studies have demonstrated that components of SEC, including AFF4, ENL, AF9 and Cyclin T1, undergo phase separation to form liquid-like droplets to enforce CTD hyperphosphorylation and to promote robust elongation of Pol II on cellular genes[6–9]. However, it is still unclear whether phase separation of SEC is involved in Tat transactivation of HIV-1 transcription during active viral replication, and if so, what molecular mechanisms are responsible for terminating this event to establish and maintain proviral latency.

The transcriptional activity of the HIV-1 Tat protein is tightly regulated to ensure proper viral replication or the establishment of proviral latency in infected cells. Although Tat is primarily localized in the nucleus, it contains a highly basic region that serves as a nucleolar localization signal and interacts with the nucleolar protein NPM1/B23, leading to its sequestration in the nucleoli[10,11]. Perturbation of Tat's intranuclear distribution impairs its transcriptional activity and HIV-1 replication, as demonstrated by the forced nucleolar localization of Tat using a nucleolus-targeted TAR decoy[12]. However, the underlying mechanisms that regulate Tat's nucleoplasmic and nucleolar trafficking and how its nucleolar compartmentalization inhibits its transactivator function remain to be elucidated. In addition to the regulation by its subcellular localization, Tat's trans-acting function is also regulated by multiple post-translational modifications, including methylation, acetylation, a phosphorylation, and ubiquitination[13]. Among these post-translational modifications, PRMT6-mediated arginine methylation negatively regulates Tat's function in viral gene transactivation by reducing its association with both TAR and P-TEFb[14,15]. These combined findings from earlier studies suggested that both the subnuclear localization and post-translational modifications of Tat are critical mechanisms for regulating its transcriptional activity. However, whether these two mechanisms operate independently or are interconnected with each other to control the transactivator function of Tat is yet to be clarified.

Protein arginine methylation is a critical post-translational modification that plays a vital role in transcriptional regulation. This modification is catalyzed by a family of enzymes called protein arginine methyltransferases (PRMTs), which consist of nine members ranging from PRMT1 to PRMT9. PRMTs can be classified into three classes based on the structure of the methylated arginine. Type I PRMTs (PRMT1, PRMT2, PRMT3, PRMT4, PRMT6, and PRMT8) catalyze monomethylarginine (MMA) and asymmetric dimethylarginine (aDMA). Type II PRMTs (PRMT5 and PRMT9) produce symmetric dimethylarginine (sDMA) and MMA. Finally, type III PRMT (PRMT7) only generates monomethylarginine (MMA)[16]. Among the PRMTs, PRMT2 stands out due to the presence of a Src homology 3 domain at its amino terminus, in addition to the well-conserved catalytic domain present in all PRMTs[17]. By catalyzing H3R8me2a and methylating other non-histone substrates, PRMT2 is implicated in many cellular processes, and deregulation of PRMT2 is closely associated with tumor development[18,19]. Although a few cellular proteins have been identified as substrates methylated by PRMT2, our understanding of the molecular details of how arginine methylation by PRMT2 affects substrate function remains limited. Additionally, little is known about the function and molecular mechanisms of PRMT2 in the context of infectious diseases.

In this study, we identify PRMT2 as a novel host restriction factor for HIV-1 transcription and proviral reactivation in both cell line and primary CD4+ T cell-based latency models, as well as in CD4+ T cells from ART-suppressed individuals. Tat physically associates with PRMT2 and is preferentially methylated at the R52 residue both in vitro and in latently infected cells. We show that Tat is targeted into the nucleoplasmic SEC liquid droplets by AFF4 phase separation to transactivate HIV-1 transcription. However, methylation of Tat by PRMT2 enhances its association with the nucleolar protein NPM1 and causes its nucleolar sequestration, which prevents AFF4-mediated Tat incorporation into the SEC droplets to promote transcription silencing and proviral latency. Overall, these results demonstrate that arginine methylation of Tat by PRMT2 controls its nucleolar localization and strongly inhibits HIV-1 transcription by impacting the formation of nucleoplasmic Tat-SEC droplets. Targeting or inhibiting PRMT2 methylase activity to promote the inclusion of Tat in the SEC droplets may represent a novel strategy for inducing viral cytopathy or immune-mediated eradication by exposing the hidden HIV-1 provirus.

## Results

### Identification of PRMT2 as a potent suppressor for Tat transactivation of HIV-1 promoter

Given the critical roles of arginine methylation of histone and non-histone proteins in the transcriptional regulation of cellular genes, we investigated the effects of pharmacological inhibition of protein arginine methylation on basal and Tat-induced transcriptional activity of the HIV-1 promoter. Blocking the methylase activity of type I PRMTs by MS023 attenuates Tat transactivation of HIV-1 promoter while the basal activity of HIV-1 promoter remains largely unchanged (Fig. 1a and Supplementary Fig. 1a). Inhibition of PRMT5, a type II PRMT, by HLCL-61, marginally affects the HIV-1 promoter activity irrespective of Tat expression (Fig. 1a and Supplementary Fig. 1a), suggesting asymmetrical arginine dimethylation by type I PRMTs could suppress Tat transactivation of HIV-1 promoter for viral gene expression. To search for PRMTs that could attenuate Tat transactivation of HIV-1 promoter, we conducted a focused expression screening using PRMTs cDNAs (Fig. 1b) and found that, similar to the effect of PAF1 expression we and other previously reported[20,21], expression of PRMT2 and PRMT6 leads to a pronounced reduction in the Tat-dependent transcriptional activity of HIV-1 promoter while the expression of other PRMTs has no or only a minor effect (Fig. 1c). The basal activity of HIV-1 promoter remains largely unaffected upon individual PRMT expression (Supplementary Fig. 1b), further suggesting that protein arginine methylation by PRMTs predominantly regulates Tat-transactivated, but not the basal activity of HIV-1 promoter. As PRMT6 demonstrated weaker activity than PRMT2 and has previously been shown to regulate HIV-1 gene expression[14], we focused our subsequent studies on investigating

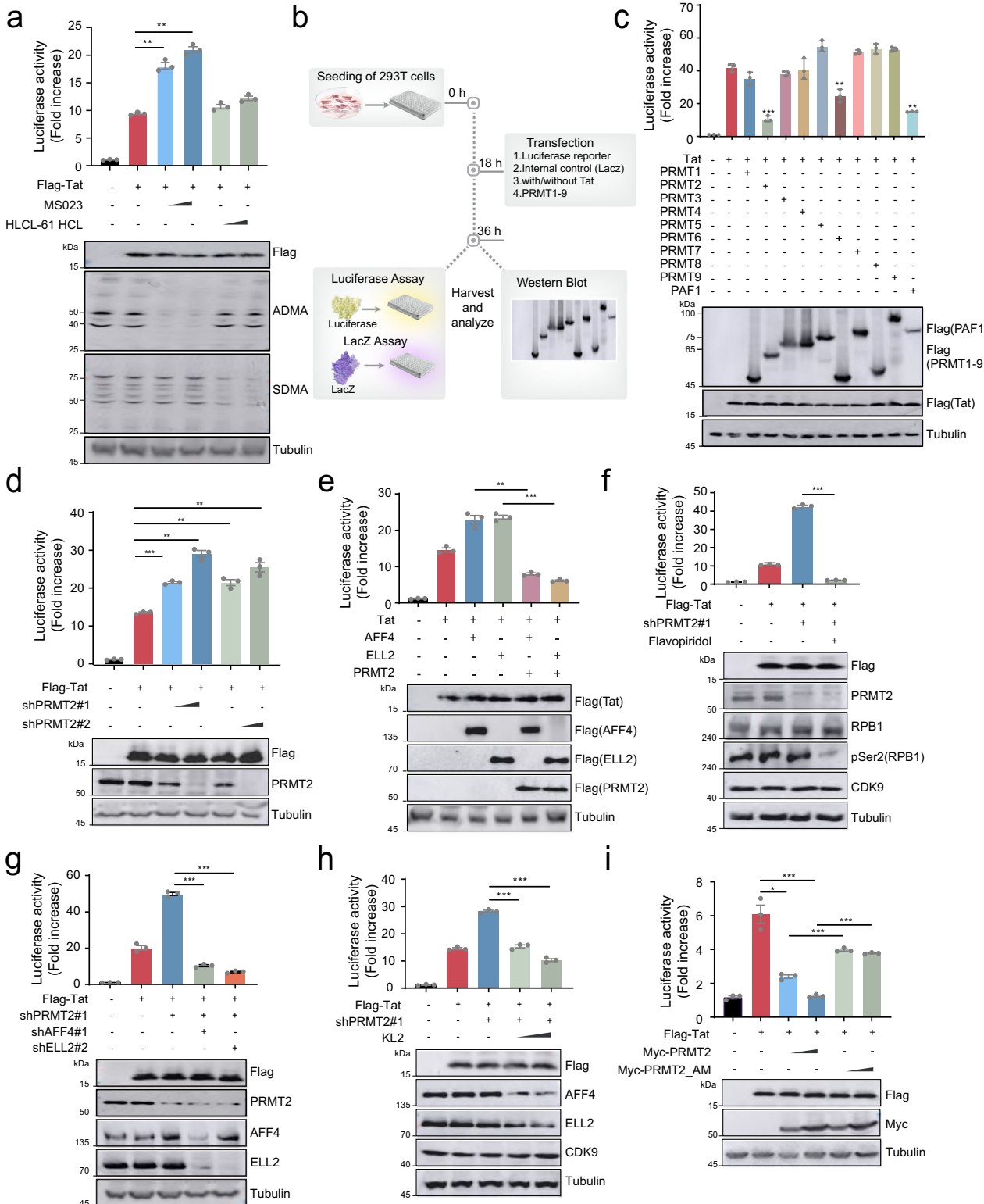

**Fig. 1 | PRMT2 suppresses Tat-dependent HIV-1 transactivation. a** Tat-induced HIV-1 LTR activity in 293T cells treated for 6 h with PRMT inhibitors. **b** Experimental outline for cDNA expression screening of PRMTs regulating HIV-1 LTR activity. **c** Tat-induced HIV-1 LTR activity in 293T cells exogenously expressing the indicated PRMT family member for 24 h. **d** Tat-induced HIV-1 LTR activity in 293T cells depleted for PRMT2 expression by two-independent lentiviral shRNAs. **e** Tat-induced HIV-1 LTR activity in 293T cells ectopically expressing the SEC subunit AFF4 or ELL2 alone or together with PRMT2. **f** Tat-induced HIV-1 LTR activity in 293T cells depleted for PRMT2 expression and treated with flavopiridol for 6 h. **g** Tat-induced HIV-1 LTR activity in 293T cells depleted for expression of PRMT2

alone or together with the SEC subunit AFF4 or ELL2. **h** Tat-induced HIV-1 LTR activity in 293T cells transduced with PRMT2 shRNAs for 48 h followed by a further 6 h of KL2 treatment. **i** Tat-induced HIV-1 LTR activity in 293T cells exogenously expressing wild-type or substrate arginine binding-deficient mutant (referred to as AM) PRMT2. Quantification in (**a** and **c–i**) was shown as mean ± SEM from three biological replicates. All immunoblots are representative of two independent experiments. Statistical significance was determined using a two-tailed Student's *t*-test (**a**, **c–i**). *$p < 0.05$; **$p < 0.01$; ***$p < 0.001$. Source data are provided as a Source Data file.

the molecular bases of PRMT2 in Tat transactivation of HIV-1 transcription, as well as its regulatory function in post-integrated proviral latency and reactivation.

To further investigate the role of PRMT2 in HIV-1 gene expression, we depleted PRMT2 and observed a significant increase in Tat-mediated transcriptional activity, but no notable change in basal HIV-1 promoter activity (Fig. 1d and Supplementary Fig. 1c). Tat is known to transactivate HIV-1 gene expression through recruiting the Super Elongation Complex (SEC) to stimulate transcriptional elongation of viral genes[3,4]. PRMT2 expression abrogates the synergistic effect of Tat and SEC on the activation of HIV-1 promoter (Fig. 1e). Conversely, inhibition of the P-TEFb kinase activity by flavopiridol, depleting the subunits AFF4 and ELL2 or disrupting the integrity of SEC using KL-2, all abrogate the increased Tat-mediated transactivation of HIV-1 promoter observed upon PRMT2 perturbation (Fig. 1f–h), suggesting that PRMT2 loss activates HIV-1 transcription likely at steps upstream of Tat-SEC's function in promoting RNA Pol II escape from HIV-1 promoter-proximal pausing.

PRMT2 carries a canonical methylation module with two pairs of glutamic acid residues for binding to SAM and substrate arginine, respectively (Supplementary Fig. 1d). Substitution of each pair of glutamate residues with glutamine completely abolishes PRMT2 methylase activity[17] and abolishes the inhibitory effect of PRMT2 expression on Tat transactivation of HIV-1 promoter (Fig. 1i and Supplementary Fig 1e). While HIV-1 transcription can be stimulated by other mechanisms such as BRD4 inhibition and PKC activation[22,23], expression of either wild-type or methylase-deficient PRMT2 does not have any apparent effects on the transcriptional activity of HIV-1 promoter induced by both BRD4 inhibition and PKC activation (Supplementary Fig. 1f, g). Together, these findings further indicate that PRMT2 suppresses HIV-1 transcription in a Tat-dependent but other mechanisms-independent manner.

## PRMT2 suppresses viral transcription to establish and sustain HIV-1 latency

As Tat acts as a potent transactivator for HIV-1 transcription and its activity is tightly linked to the post-integrated proviral infection state[2], we speculated that PRMT2 might play a role in the regulation of proviral latency and reactivation. To test this possibility, we depleted PRMT2 in two Jurkat-derived E4 and 2D10 clonal cell lines that are integrated with a single copy of latent HIV-1 provirus at *CENPP* and *MSRB1* loci, respectively[24] (Fig. 2a and Supplementary 2a, b). In both clonal cell lines, we observed that PRMT2 depletion leads to a time-dependent increase in proviral transcription and reversal of latency, reaching up to -20% of E4 and -12% of 2D10 cells 9 days post PRMT2 depletion (Fig. 2b, c and Supplementary Fig. 2c, d). Furthermore, PRMT2 depletion also facilitates transcriptional reactivation and latency reversal of the HIV-1 provirus in both E4 and 2D10 cells induced by LRAs, TNFa, PMA and JQ1, respectively (Fig. 2d, e and Supplementary Fig. 2e–i), indicating that PRMT2 suppresses viral transcription to both maintain proviral latency and antagonize LRAs-induced latency reversal.

Given the methylase activity of PRMT2 is critical for inhibiting Tat transactivation of HIV-1 promoter, we wondered if PRMT2 also employs this activity to maintain proviral latency and counteract LRAs-induced latency reversal. To address this question, we introduced exogenous wild-type and methylase-deficient PRMT2 into mixed polyclonal E4 cells that had been knocked out for PRMT2 by CRISPR/Cas9 (Fig. 2f). Flow cytometry analyses showed that the expression of wild-type PRMT2, but not the methylase-deficient variant, potently suppresses the proviral reactivation triggered by PRMT2 ablation (Fig. 2g). Moreover, the introduction of exogenous wild-type PRMT2 effectively antagonizes the latency reversal in both control and PRMT2-knockout E4 cells (Fig. 2h). Conversely, this effect was not observed when exogenous methylase-deficient PRMT2 was re-introduced

(Fig. 2g, h), thus establishing the methylase activity as an essential mechanism for sustaining latency and counteracting proviral reactivation following LRAs treatment.

Upon removal of the activation signal, reactivated HIV-1 proviruses in latently infected cells undergo rapid reversion to a transcriptionally inactive state. This event necessitates the disruption of the Tat-mediated positive regulatory circuit to abolish processive Pol II elongation on viral gene[24]. Thus, we reasoned that PRMT2 might function as a mediator or accelerator for the re-entry of proviral latency. To test this notion, we stimulated 2D10 cells with TNFα overnight to maximally reactivate proviral transcription to comparable levels in cells with or without PRMT2 depletion and then compared the kinetics of reversion to proviral latency. Flow cytometry analyses of the proportion of GFP+ cells and quantification of d2GFP and env transcripts downstream of HIV-1 LTR reveal a markedly slower shutdown kinetics of proviral transcription in PRMT2-depleted cells as compared with the control cells (Fig. 2i and Supplementary Fig. 2j). In contrast, introduction of wild-type PRMT2 into 2D10 cells leads to a reduced frequency of GFP+ cells and an accelerated shutdown kinetics of proviral transcription, a phenomenon not reproduced with expression of the methylase-deficient counterpart (Fig. 2j and Supplementary Fig. 2k, l). Together, these data indicated that PRMT2 is also essential for the transcriptional silencing of viral gene for the establishment of HIV-1 latency.

## PRMT2 promotes HIV-1 latency in primary CD4+ T cells of healthy donors and HIV-1-infected patients

To investigate the role of PRMT2 in the establishment of viral latency during initial HIV-1 infection, we utilized a dual-color HIV-1$_{GKO}$ reporter virus, in which the expression of codon-switched EGFP (csGFP) and mKO2 fluorescent proteins are under the control of HIV-1 LTR and the constitutively active elongation factor 1α (EF1α) promoter, respectively (Fig. 3a). In this reporter system, cells infected with HIV-1$_{GKO}$ dual-labeled virus express mKO2 fluorescent protein, with productively and latently infected cells marked by the presence and absence of GFP expression, respectively. Using this system, we depleted PRMT2 in Jurkat T cells and then infected these cells with the HIV-1$_{GKO}$ virus. Despite the comparable infection rates in both control and PRMT2-depleted cells, flow cytometry results demonstrated that PRMT2 depletion leads to a prominent increase in actively infected cells and a concomitant reduction in the proportion of latently infected cells (Fig. 3b). Conversely, introduction of wild-type PRMT2 markedly increased the proportion of latent infection in Jurkat T cells transduced with HIV-1$_{GKO}$ virus while the proportions of latently and actively infected cells are largely unaffected by the expression of a methylase-inactive PRMT2 (Fig. 3c), indicating that PRMT2 promotes latency establishment during viral infection in a methylase-dependent manner.

The latent HIV-1 proviruses predominantly reside in the resting CD4+ T cells, which was found to be established either directly during the early stage of infection of the quiescent CD4+ T cells or from the infection of the activated CD4+ T cells that later transit to a resting state[25]. Many cellular factors involved in the HIV-1 replication, such as P-TEFb and NF-kB, are differentially expressed or regulated in the resting and activated CD4+ T cells to mediate a switch between proviral latency and productive replication[26,27]. To explore whether PRMT2 also regulates HIV-1 latency in a pathophysiological setting, we first examined its expression profile in the resting and activated CD4+ T cell using the Genomic Utility for Association and Viral Analyses in HIV (GuavaH) database[28] and found that PRMT2 expression rapidly decreases upon the in vitro activation of resting primary CD4+ T cells isolated from healthy donors and HIV-1 patients on antiretroviral therapy (ART) (Fig. 3d). This observation was validated in the resting primary CD4+ T cells of two healthy donors upon in vitro activation using T cell receptors (TCRs) engagement with anti-CD3/CD28 antibodies or by

 

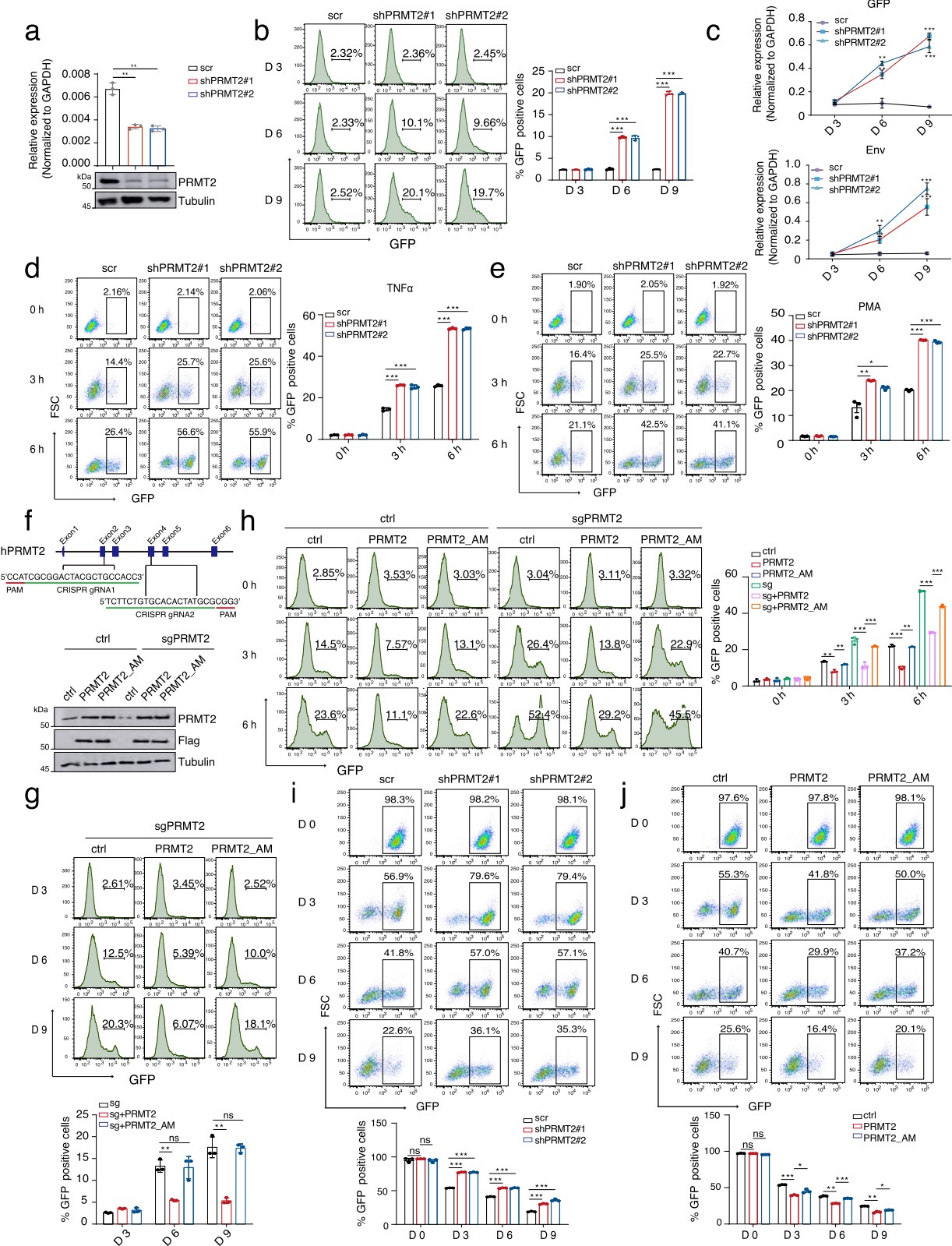

protein Kinase C (PKC) activation with PMA (Fig. 3e and Supplementary Fig. 3a), indicating the potential involvement of PRMT2 in HIV-1 latency regulation in the primary T cells. Flow cytometry analyses of the HIV-1$_{GKO}$ viral infection of activated primary CD4$^+$ T cells from two anonymous healthy donors demonstrated that depletion of PRMT2 leads to a marked increase in the proportion of productively infected cells and a concomitant significant decrease in latent infection (Fig. 3f,

g). On the contrary, exogenous expression of wild-type PRMT2, rather than a methylase-deficient variant, markedly increases the latent infection of primary CD4$^+$ T cells by HIV-1$_{GKO}$ viruses (Supplementary Fig. 3b). We further examined the regulatory function of PRMT2 in the reactivation of latent provirus in primary cells by infecting CD4$^+$ T cells with luciferase gene-containing HIV-1 reporter viruses (Fig. 3h). Exogenous expression of wild-type PRMT2 strongly suppresses the

**Fig. 2 | PRMT2 restricts viral transcription to promote the establishment and maintenance of HIV-1 latency in a methyltransferase-dependent manner. a** RT-qPCR and immunoblotting analyses of PRMT2 levels in E4 cells 72 h post lentiviral transduction with two independent shRNAs. **b** Proportion of GFP+ cells analyzed by flow cytometry at indicated days post PRMT2 depletion in E4 cells. **c** Quantification of GFP (upper panel) and HIV env (lower panel) mRNA levels in E4 cells in (**b**). **d**, **e** Proportion analyses of GFP+ cells by flow cytometry in control and PRMT2-depleted E4 cells stimulated with TNFα (**d**) or PMA (**e**) for indicated times. **f** Scheme for PRMT2 knockout by CRISPR/Cas9 (upper) and immunoblotting analyses of the indicated protein in control and PRMT2-knockout polyclonal E4 cells rescued with empty vector, wild-type PRMT2 or its AM mutant (lower). **g** E4 cells were treated as in (**f**). The proportion of GFP+ cells was determined by flow cytometry at indicated days. **h** E4 cells were treated as in (**f**) and the proportion of GFP+ cells at indicated

times following TNFα stimulation was determined by flow cytometry. **i** Representative flow pseudocolor plots showing the frequency of the GFP+ population in 2D10 cells that were fully reactivated by TNFα overnight followed by its washout and continued culturing for indicated times. **j** Representative flow pseudocolor plots showing GFP+ population at indicated days following TNFα washout in control, wild-type, or AM PRMT2-expressing 2D10 cells that were fully reactivated by TNFα stimulation overnight. Quantifications in (**a–e, g** and **h**) are shown as mean ± SD from three biological replicates, Quantifications in (**i** and **j**) are shown as mean ± SEM from three biological replicates. All western blots are representative of three independent experiments. Statistical significance was determined using a two-tailed Student's $t$-test (**a–e, g–j**). *$p < 0.05$; **$p < 0.01$; ***$p < 0.001$. Source data are provided as a Source Data file.

proviral reactivation induced by both TCR engagement and PKC activation, which is fully dependent on its methylase activity (Fig. 3i and Supplementary Fig. 3c).

To investigate the impact of PRMT2 depletion on HIV-1 latency in a clinically relevant context, we isolated primary CD4+ T cells from the PBMCs of four HIV-1-infected patients who were on suppressive ARTs for at least two years, with undetectable viral loads (<50 copies/ml) and high CD4+ T cell counts (>500 cells/ml). Following isolation, the cells were transduced with lentivirus encoding control and PRMT2-specific shRNAs and stimulated with LRAs for three days, after which total RNA was extracted to assess proviral reactivation (Fig. 3j). RT-qPCR analysis of LTR transcript levels revealed that PRMT2 depletion alone induced a mild but statistically significant increase in HIV-1 viral gene expression. Additionally, while existing LRAs exhibited varying degrees of HIV-1 reactivation, depletion of PRMT2 significantly enhanced the latency-reversing effects of all LRAs tested in these patient cells (Fig. 3k). In summary, these results demonstrated that PRMT2 promotes proviral latency and its depletion can cooperate with existing LRAs to reactivate HIV-1 ex vivo in patient-derived primary CD4+ T cells.

## PRMT2 associates with Tat and attenuates its binding to the SEC

To decipher the underlying mechanisms by which PRMT2 suppresses Tat-mediated activation of HIV-1 transcription, Tat-associated proteins were immuno-purified from 293T cells and subjected to mass spectrometric analyses. In addition to the known interactors, including the NPM1 and components of the SEC[4,11], several peptides of PMRT2 were also identified, pointing to an interaction between PRMT2 and Tat (Fig. 4a). Co-immunoprecipitation experiments confirmed their association in 293T cells, the latently infected E4 cells after reactivation and in primary CD4+ T cells infected with HIV-1$_{GKO}$ reporter virus (Fig. 4b, c and Supplementary Fig. 4a, b). Gradient centrifugation using nuclear extracts of Tat-expressing 293T cells showed the co-migration of endogenous PRMT2 and Tat, with the highest overlap from fractions 13 to 15 (Supplementary Fig. 4c). Moreover, the direct interaction between Tat and PRMT2 was demonstrated in a GST pull-down assay using recombinant PRMT2 and Tat expressed and purified from bacteria (Fig. 4d). PRMT2 harbors a characteristic arginine methylase module at its C-terminus and an N-terminal Src homology 3 (SH3) domain (Supplementary Fig. 4d). Co-immunoprecitation and GST pull-down using the full-length PRMT2 and its truncating mutants revealed that the N-terminal SH3-containing domain is essential and sufficient for its binding to Tat (Fig. 4e and Supplementary Fig. 4e), which is analogous to the mechanism used by PRMT2 and PRMT4/CARM1 to recognize and associate with their substrates[29,30].

The luciferase reporter assay using the full-length PRMT2 and truncating mutants demonstrated that either deletion of the C-terminal catalytic module or the N-terminal SH3 domain abolishes the suppressive effect of PRMT2 on the Tat transactivation of HIV-1 LTR, highlighting both the arginine methylase and Tat-binding activities are essential for its inhibitory function in HIV-1 transcription (Fig. 4f). To characterize the underlying molecular mechanisms, we

assessed the association of Tat with the SEC components, and found that PRMT2 expression potently suppress the Tat-containing SEC formation in both 293T cells and E4 cells, which largely relies on both its methylase and Tat binding activities (Fig. 4g, h, and Supplementary Fig. 4f). Conversely, PRMT2 depletion promotes Tat association with the SEC subunits in E4 cells after proviral reactivation and primary CD4+ T cells infected with HIV-1$_{GKO}$ reporter viruses (Fig. 4i and Supplementary Fig. 4g). Tat exerts its transactivating function in HIV-1 expression primarily through recruiting the SEC to promote the release of paused RNA Pol II from HIV-1 promoter[3,31]. Consistent with the above interaction findings, chromatin immunoprecipitation followed by quantitative PCR (ChIP-qPCR) analyses revealed a much higher abundance of Tat and the SEC components, such as the Cyclin T1, AFF4 and ELL2, on the HIV-1 promoter, as well as an increased level of elongating RNA Pol II across proviral genes in PRMT2-depleted E4 cells after proviral reactivation as compared to the control cells (Fig. 4j, k and Supplementary Fig. 4h). In contrast, the ectopic expression of wild-type PRMT2, but not a methylase-inactive or Tat-binding activity-deficient variant impairs the recruitment of Tat and the SEC to the HIV-1 LTR, leading to a pronounced reduction of elongating RNA Pol II throughout proviral gene (Fig. 4l, m and Supplementary Fig. 4i).

## PRMT2 preferentially methylates Tat at arginine 52 upon re-entry into proviral latency

The Tat protein is highly variable among different HIV-1 clades and can tolerate up to 40% sequence variation without any noticeable loss of transcriptional activity[32]. Sequence alignment of 180 full-length Tat proteins revealed the absolute conservation of arginine residues at positions 49, 52, 53, 55, 56, and relatively conserved residues at 57 and 78 (Fig. 5a). Mutational analyses demonstrated that substitution of arginine 52 and 53 with lysine or alanine attenuates Tat transactivation of HIV-1 promoter in a synergistic manner, while substitution of other arginine residues has no visible effect on Tat transcriptional activity (Fig. 5b and Supplementary Fig. 5a). Furthermore, ectopic expression of PRMT2 continues to suppress the transcriptional activity of Tat with lysine replacement at most arginine residues except for R52K mutant, even though it shows the reduced potency to transactivate HIV-1 promoter (Fig. 5c and Supplementary Fig. 5b), suggesting that PRMT2 attenuates the Tat transcriptional activity in a R52-dependent manner.

R52 is located within the basic RNA-recognition motif (amino acid 48 to 57) that is essential for the nuclear import of Tat and its binding to the TAR[33]. Previous studies have reported that several lysine and arginine residues within this motif, including R52, are methylated to regulate Tat's transcriptional activity[15,34]. Thus, we speculated that Tat may act as a substrate for arginine methylation by PRMT2. To determine whether PRMT2 can methylate Tat, we performed in vitro methyltransferase assays by incubating recombinant his-Tat and $^3$H-AdoMet with wild-type or methylase-inactive PRMT2 that were affinity-purified from 293T cells (Supplementary Fig. 5c, d). Liquid scintillation counting of the protein band representing Tat on SDS-PAGE gel revealed high level of $^3$H-methyl

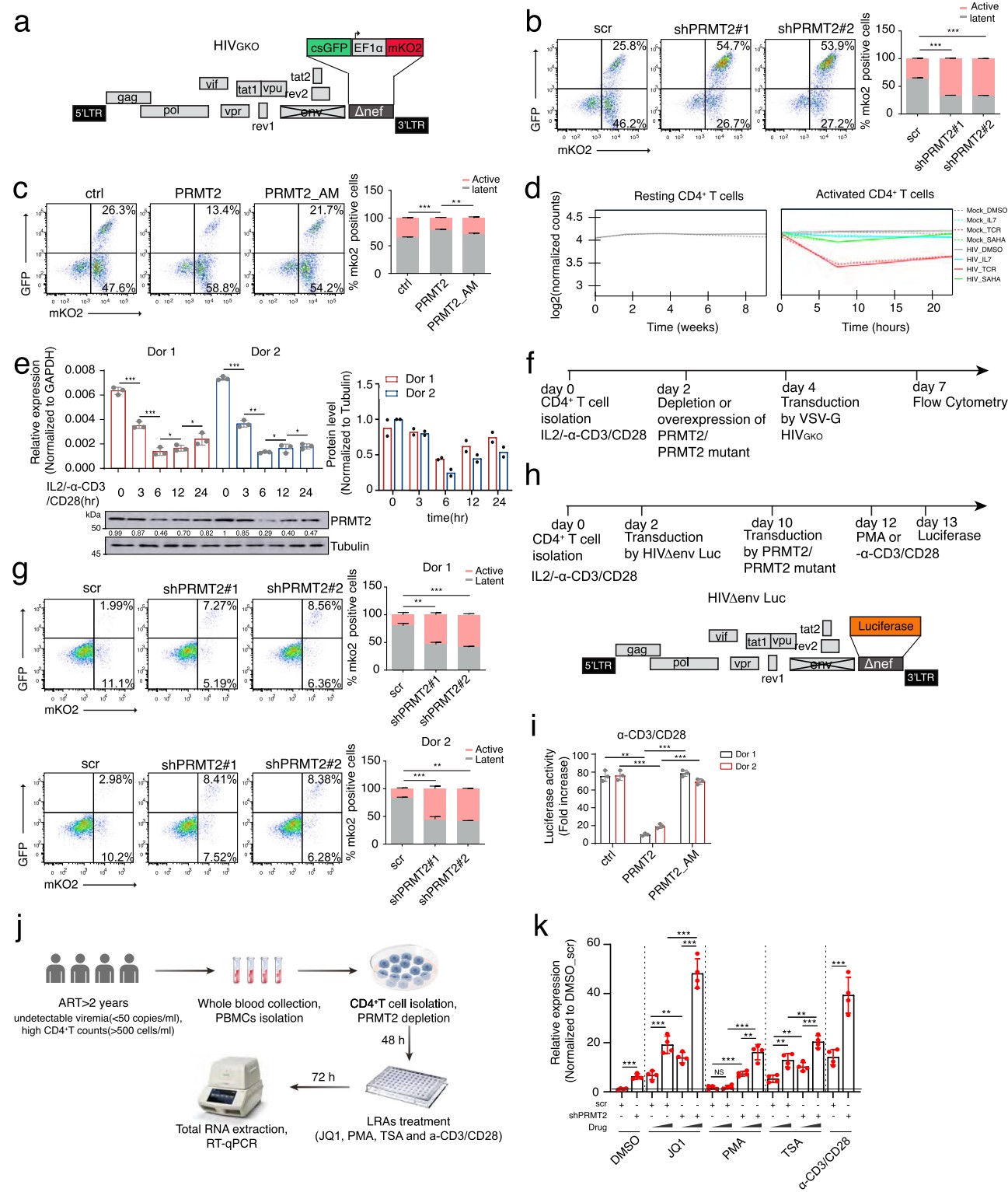

incorporation into Tat protein by wild-type PRMT2 incubation, whereas the two methylase-inactive PRMT2 mutants do not manifest any notable activity (Fig. 5d). Additionally, ectopic expression of PRMT2 leads to a prominent increase in the level of methylated Tat, while CRISPR/Cas9-mediated knockout causes a notable reduction (Fig. 5e, f and 2f). Intriguingly, though ectopic expression or depletion of PRMT6, another known Tat arginine methylase[14], only displays mild effects on Tat methylation, its ablation further decreases the level of methylated Tat in PRMT2-ablated cells (Fig. 5e, f and Supplementary Fig. 5e), indicating that PRMT2 is a primary arginine

methylase for Tat and can methylate Tat with PRMT6 in a synergistic fashion.

We next sought to determine the methylation status of R52 and R53 in Tat protein by PRMT2. Co-expression of wild-type or mutant Tat with PRMT2 in 293T cells revealed a significant decrease in arginine methylation levels in R52K and to a lesser extent in R53K mutants, and concurrent mutation of R52 and R53 abolished PRMT2-induced Tat arginine methylation (Fig. 5g). In vitro methylation assay using affinity-purified PRMT2 and wild-type or mutant recombinant Tat further confirmed that PRMT2 primarily methylated R52 and to a lesser extent

**Fig. 3 | PRMT2 promotes HIV-1 latency in primary CD4+ T cells of healthy donors and HIV-1-infected patients. a** Schematic representation of dual-labeled fluorescent HIV-1GKO reporter. **b** Representative flow pseudocolor plots of control and PRMT2-depleted Jurkat T cells infected with HIV-1GKO reporter lentivirus for 72 h. **c** Representative flow pseudocolor plots of Jurkat T cells transduced with indicated expression construct followed by lentiviral HIV-1GKO reporter infection for 72 h. **d** Temporal expression pattern of PRMT2 mRNA in normal and HIV-1 infected primary CD4+ T cells during resting and activation stages induced by various stimulators. **e** RT-qPCR and immunoblotting analyses of PRMT2 level in primary CD4+ T cells either left untreated or activated with IL-2 plus anti-CD3/CD28 antibodies for indicated times. **f** Experimental schedule for VSV-G pseudotyped HIV-1GKO virus infection of primary CD4+ T cells following endogenous PRMT2 depletion by shRNAs or ectopic expression of wild-type or methylase-inactive PRMT2. **g** Flow cytometry analyses of primary CD4+ T cells infected with HIVGKO virus following PRMT2 depletion as outlined in (**f**). **h** Schematic representation of HIV-1NL4-3 luciferase reporter and experimental procedure for infection of primary CD4+ T cells from healthy donors followed by proviral reactivation and luciferase analyses. **i** Luciferase activity in primary CD4+ T cells transduced with HIV-1NL4-3 reporter virus outlined in (**h**) followed by reactivation with anti-CD3/CD28 for 20 h. **j** Experimental design for reactivation of provirus with LRAs in primary CD4+ T cells from ART-suppressed HIV-1-infected individuals. **k** HIV-1 LTR level in patient-deprived CD4+ T cells treated as in (**j**) was determined by RT-qPCR. Quantifications in (**b, c, e, g** and **i**, $n = 3$) and in (**k**, $n = 4$) are shown as mean ± SD. Values from densitometric analyses of immunoblots are shown beneath the respective gel bands and quantification of protein levels in (**e**) is shown as mean ($n = 2$). Statistical significance was determined using a two-tailed Student's *t*-test (**b, c, e, g, i**, and **k**). *$p < 0.05$; **$p < 0.01$; ***$p < 0.001$. Source data are provided as a Source Data file.

R53 (Fig. 5h and Supplementary Fig. 5f). To examine whether Tat methylation occurs in HIV-1 latently infected cells, we generated two rabbit polyclonal antibodies against R52ame2 Tat. Dot blot analyses showed that both antibodies specifically recognize synthetic R52ame2-containing but not unmodified corresponding peptides (Supplementary Fig. 5g). Furthermore, one of the antibodies (TatR52ame2_2) can also recognize R52ame2 of ectopically expressed Tat in 293T cells co-expressing PRMT2 while another one (TatR52ame2_1) loses its ability to recognize the cellular Tat R52ame2 irrespective of PRMT2 expression (Supplementary Fig. 5h). Thus, the TatR52ame2_2 antibody was utilized to probe cellular Tat methylation in the following studies. In line with the results observed with ADMA antibody, ectopic expression of PRMT2 and PRMT6 markedly increases the TatR52ame2 level while expression of methylase-defective PRMT2 has no detectable effect (Supplementary Fig. 5i). Knockouts of PRMT2 and PRMT6 in 293T cells also led to a prominent reduction of TatR52ame2 level (Supplementary Fig. 5j). Immunoblotting analyses using TatR52ame2_2 antibody revealed that R52 methylation was completely undetectable when Tat was mutated at R52 alone or in combination with R53 (Supplementary Fig. 5k). Together, these results confirm that PRMT2 asymmetrically dimethylates ectopically expressed Tat on R52, which could be specifically detected with our homemade TatR52ame2_2 antibody.

To determine whether R52ame2 modification on Tat also occurs in HIV-1 latently infected cells, we conducted a detailed analysis of methylation kinetics in 2D10 cells that are stimulated with TNFα for proviral reactivation and then washed out to allow the provirus to re-enter a latent state. Immunoblotting analyses with both ADMA and TatR52ame2_2 antibodies revealed that the TatR52ame2 level is initially low during proviral reactivation, but gradually increases during the reestablishment of proviral latency (Fig. 5i). PRMT2 depletion prevents asymmetric dimethylation of TatR52 during re-entry into a latent state (Fig. 5j). Moreover, PRMT2 deletion also leads to a marked reduction in Tat R52 methylation in primary CD4+ T cells infected with HIV-1GKO reporter viruses (Fig. 5k). Collectively, these results indicated that PRMT2 methylates Tat on R52 in HIV-1 latently infected cells, which primarily occurs during the latency re-entry of reactivated provirus.

### AFF4 promotes Tat nucleolar exit and its phase separation incorporates Tat into the nucleoplasmic SEC droplets

The Tat protein is a nuclear protein that exhibits frequent shuttling between the nucleoli and the nucleoplasm, and its subnuclear location is critical for its function[10]. Earlier studies have demonstrated that the physical association of Tat with a nucleolar protein, NPM1 (also known as B23), is necessary for Tat's nucleolar localization[11]. In agreement with these findings, our confocal microscopy analyses revealed that Tat is unevenly distributed in the nucleus, with multiple prominent small nuclear puncta labeled by the nucleolar marker, fibrillarin (Fig. 6a, b). Further subcellular fractionation of Hela cells exogenously expressing flag-Tat demonstrated that most Tat co-localizes with NPM1 in the nucleolar fraction, whereas the SEC components are predominantly present in the nucleoplasmic fraction (Supplementary Fig. 6a, b). Remarkably, we noticed that ectopic expression of NPM1 leads to a marked increase in Tat abundance within the nucleolus and a concomitant reduction in the level of nucleoplasmic Tat in Hela cells, whereas expression of the SEC scaffold protein, AFF4, causes nucleolar fragmentation and marked increase in the level of Tat in the nucleoplasm, resulting in the formation of a few large puncta that co-localize with AFF4 (Fig. 6a–c). These results suggest that AFF4 and NPM1 may compete for Tat binding to regulate its subcellular distribution. Consistent with this idea, ectopic expression of NPM1 in 293T cells strongly inhibits Tat's association with the SEC components, while AFF4 expression attenuates the interaction between Tat and NPM1 (Fig. 6d and Supplementary Fig. 6c). Additionally, knockdown of NPM1 promotes the formation of Tat-containing SEC complex in E4 cells undergoing proviral reactivation (Fig. 6e). Subcellular fractionation of Hela cell extracts also revealed that NPM1 expression increases the levels of nucleolar Tat, whereas ectopic expression of AFF4 shifts its distribution toward nucleoplasmic regions (Fig. 6f).

Recently, it has been reported that several subunits of the SEC, such as AFF4, CYCT1 and ENL undergo liquid-liquid phase separation to form the dynamic droplet-like condensates both in vitro and in cells[6–8]. Given the established physical association of Tat with the SEC as well as the presence of nuclear Tat puncta concurrently labeled by AFF4, we reasoned that Tat could be incorporated into the nucleoplasmic phase-separated SEC droplets via its direct interaction with the SEC constituents. To explore whether other subunits of the SEC can also undergo phase separation and incorporate Tat into their puncta, we co-expressed mCherry-Tat with individual GFP-tagged subunits of the SEC in Hela cells. However, unlike AFF4, none of the additional SEC subunits could incorporate Tat into their puncta and increase the level of Tat in the nucleoplasm, even though many of them also underwent liquid-liquid phase separation to form droplets (Supplementary Fig. 6d–f). Notably, the expression of AFF4 results in the incorporation of both Tat and other components of the SEC into its droplets, which is in line with the current understanding that AFF4 acts as a scaffold for SEC formation and directly interacts with Tat[3,4] (Fig. 6g, h).

AFF4 contains a long, intrinsically disordered region capable of forming phase-separated droplets both in vitro and in cells[6] (Supplementary Fig. 6g). Previous structural studies have shown that the amino-terminal 73 amino acid fragment of AFF4 directly binds to Tat and enhances its affinity for P-TEFb[35,36]. In light of these observations, we predicted that the amino-terminal fragment of AFF4 may mediate Tat incorporation into its droplets. To test this idea, we conducted in vitro droplet formation assays and found that recombinant mCherry-Tat did not undergo phase separation in the presence of GFP protein, but readily formed droplets when incubated with a recombinant amino-terminal 200 aa AFF4 truncating mutant (Fig. 6i, j and Supplementary Fig. 6h). Phenotypic characterization of these in vitro droplets revealed that they are spherical in shape, highly dynamic and can fuse with each other when they are in close proximity (Fig. 6k–m).

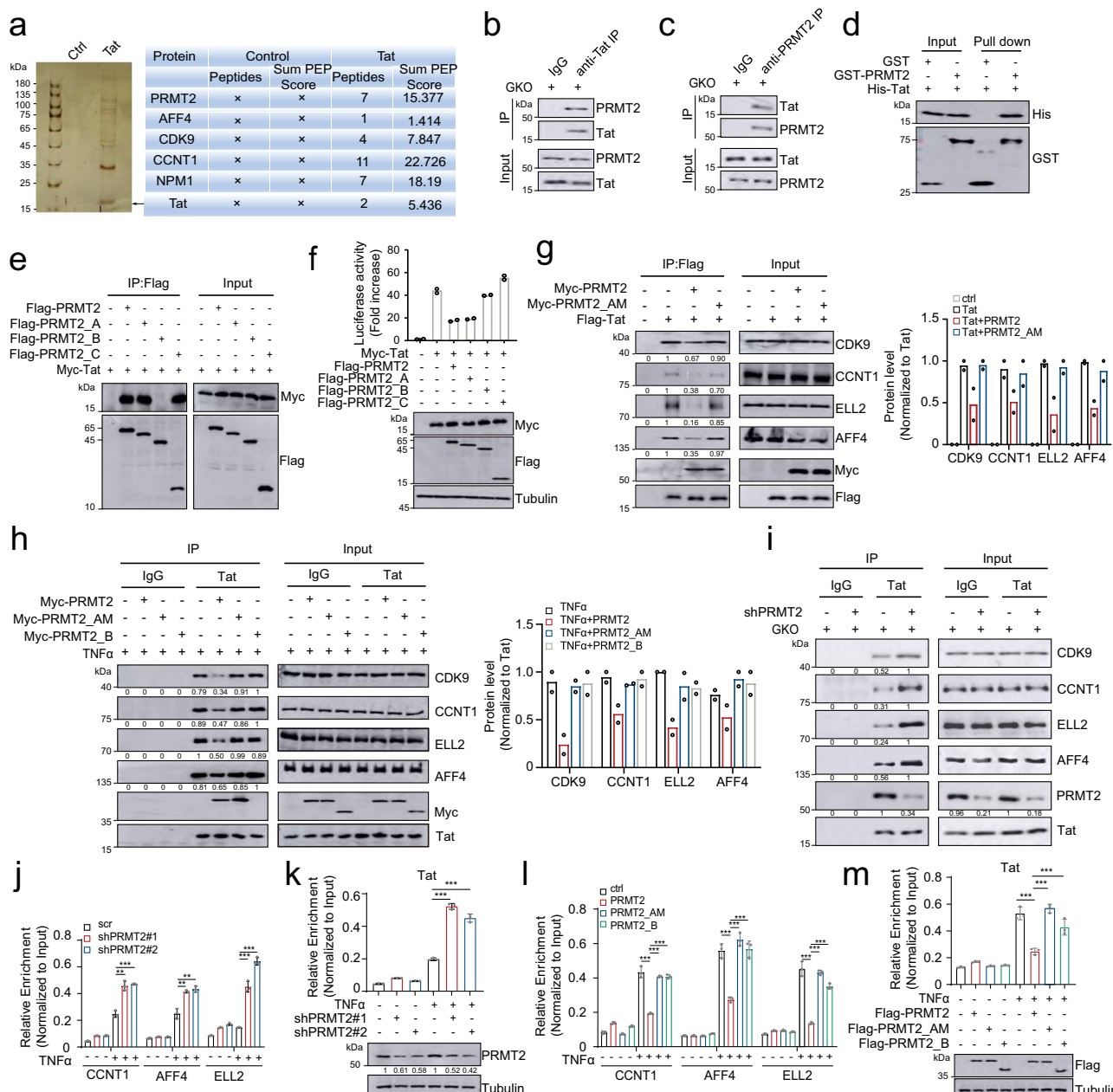

**Fig. 4 | PRMT2 attenuates the cellular level of the Tat-SEC complex and its recruitment to the HIV-1 locus via direct association with Tat. a** Silver staining visualization and mass spectrometric identification of Tat-associated proteins in 293T cells. **b, c** Primary CD4$^+$ T cells from healthy donors were infected with HIV-1$_{GKO}$ reporter virus and immunoprecipitated with anti-IgG and anti-Tat (**b**) or anti-PRMT2 (**c**) followed by immunoblotting analyses with indicated antibodies. **d** Immunoblotting analyses of indicated protein bound to the immobilized GST or GST-PRMT2 in GST pull-down assay. **e** Immunoblotting analyses of the whole cellular extract and anti-flag immunoprecipitates from 293T cells co-transfected with indicated constructs. **f** HIV-1 LTR activity in 293T cells 24 h post co-transfection with indicated constructs. **g** Nuclear extracts from 293T cells co-transfected with indicated constructs were subjected to immunoprecipitation and immunoblotting analyses. **h** E4 cells transduced with the indicated construct were

immunoprecipitated with anti-IgG or anti-Tat and then subjected to immunoblotting analyses. **i** Primary CD4$^+$ T cells with or without PRMT2 depletion were infected with HIV-1$_{GKO}$ reporter virus for 48 h before subjecting to immunoprecipitation and immunoblotting analyses. **j, k** ChIP-qPCR analyses of the levels of SEC (**j**) and Tat (**k**) on the LTR in control and PRMT2-depleted E4 cells treated with TNFα for 6 h. **l, m** ChIP-qPCR analyses of the levels of SEC (**l**) and Tat (**m**) on the LTR in E4 cells that were transduced with indicated PRMT2 and then treated with TNFα for 6 h. Quantification in (**j**–**m**, $n = 3$) and (**f**, $n = 2$) are shown as mean ± SD. Values from densitometric analyses of immunoblots are shown beneath respective gel bands and/or as mean in bar graphs (**g** and **h**, $n = 2$). Statistical significance was determined using a two-tailed Student's t-test (**j**–**m**). *$p < 0.05$; **$p < 0.01$; ***$p < 0.001$. Source data are provided as a Source Data file.

---

The crystal structure of the Tat-AFF4-P-TEFb quaternary complex revealed that the interaction between Tat and AFF4 is mediated by an interface consisting of several residues of AFF4 (E61, M62, F65 and D68) and Tat (E2, K28 and F32)[37]. To determine if this interaction is required for the Tat incorporation into AFF4 droplets, we mutated multiple interface residues of both AFF4 (E61, M62 and F65) and Tat

(K28 and F32) to alanine. GST pull-down and co-immunoprecipitation analyses showed that the physical interaction between Tat and AFF4 was completely lost in vitro, and substantially reduced in cells when either the interaction interface residues of AFF4 or Tat were mutated (Supplementary Fig. 6i–l).In vitro droplets formation assay demonstrated that the recombinant AFF4 (E61A, M62A and F65A) mutant fails

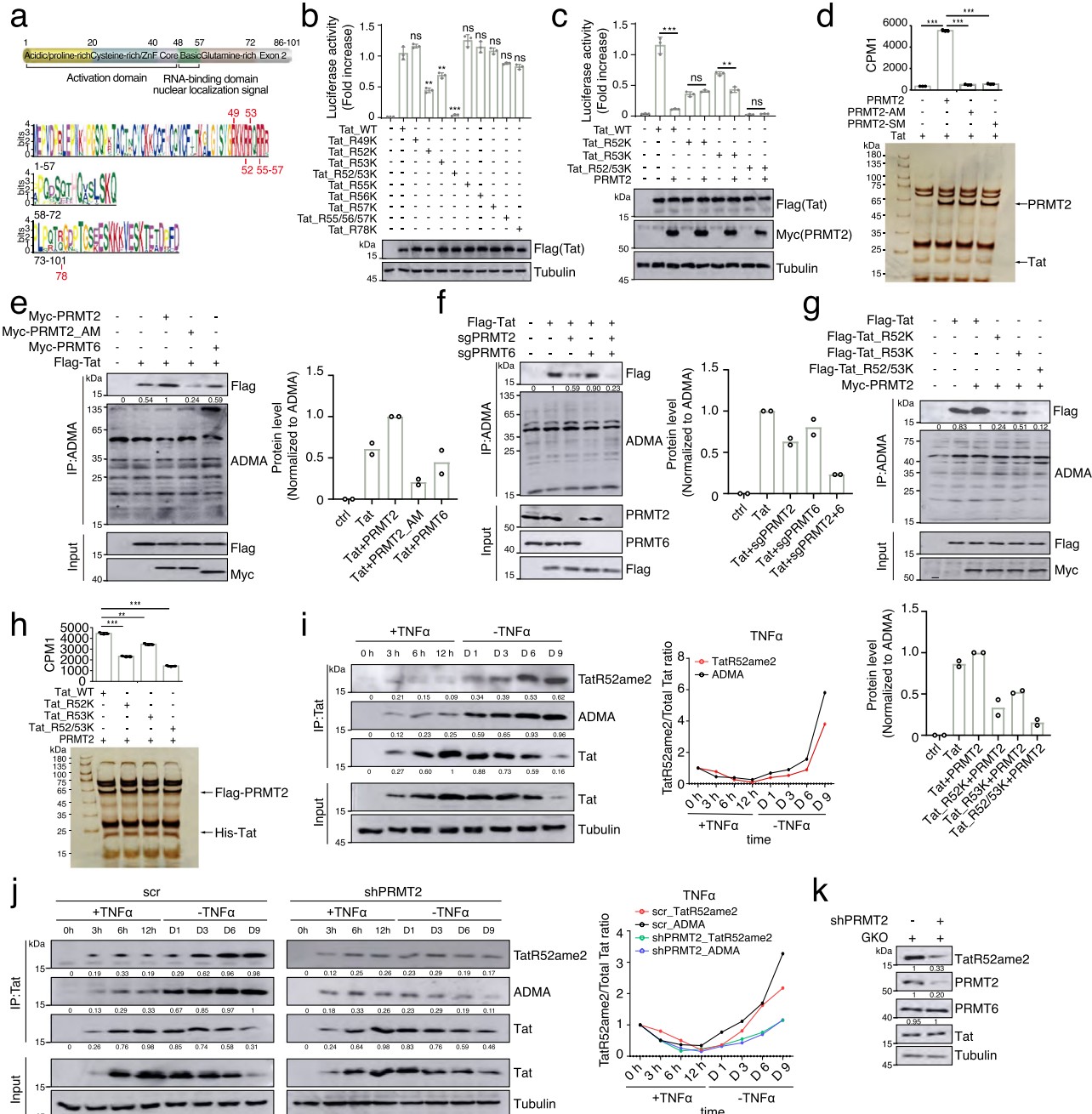

**Fig. 5 | PRMT2 preferentially methylates R52 of Tat to suppress its transactivation of HIV-1. a** Tat domain organization and sequence logos across HIV clades. **b** HIV-1 LTR activity in 293T cells expressing wild-type and indicated mutant Tat protein. **c** HIV-1 LTR activity in 293T cells 24 h post-transfection of indicated expression constructs. **d** In vitro methylation of recombinant Tat protein by affinity-purified wild-type or catalytically inactive PRMT2 in the presence of ³H-SAM. The methylation signal was determined by liquid scintillation. **e** Immunoblotting analyses of input and anti-ADMA immunoprecipitates from 293T cells co-expressing indicated proteins. **f** Immunoblotting analyses of input and anti-ADMA immunoprecipitates from 293T cells knocked out for PRMT2 or PRMT6 followed by an ectopic expression of flag-Tat. **g** Immunoblotting analyses of input and anti-ADMA immunoprecipitates from 293T cells transduced with indicated constructs. **h** In vitro methylation assay was performed as in (**d**) using wild-type Tat and its mutants as substrates. **i** The levels of the arginine-methylated and total Tat in 2D10 cells at indicated time points following the addition and

withdrawal of TNFα were analyzed by immunoblotting. The signals for arginine-methylated and total Tat were calculated by Image J and normalized to Tubulin at each time point. The ratios for the signals between arginine-methylated to total Tat were shown as fold change relative to that in the cells before TNFα addition (0 h), which is set to 1. **j** Immunoblotting analyses of the arginine-methylated and total Tat levels in control and PRMT2-depleted 2D10 cells at indicated time points following addition and withdrawal of TNFα. The levels of arginine-methylated, total Tat, and their ratios were quantified as in (**i**). **k** Primary CD4⁺ T cells of healthy donors were transduced with control or PRMT2 shRNA and then subjected to infection with HIV-1_GKO reporter virus before conducting immunoblotting analyses with indicated antibodies. Quantifications in (**b–d** and **h**, *n* = 3) are shown as mean ± SD. Values from densitometric analyses of immunoblots are shown beneath respective gel bands and/or as mean in bar graphs (**e–g**, **i** and **j**, *n* = 2). Statistical significance was determined using a two-tailed Student's *t*-test (**b–d** and **h**). *$p < 0.05$; **$p < 0.01$; ***$p < 0.001$. ***$p < 0.001$. Source data are provided as a Source Data file.

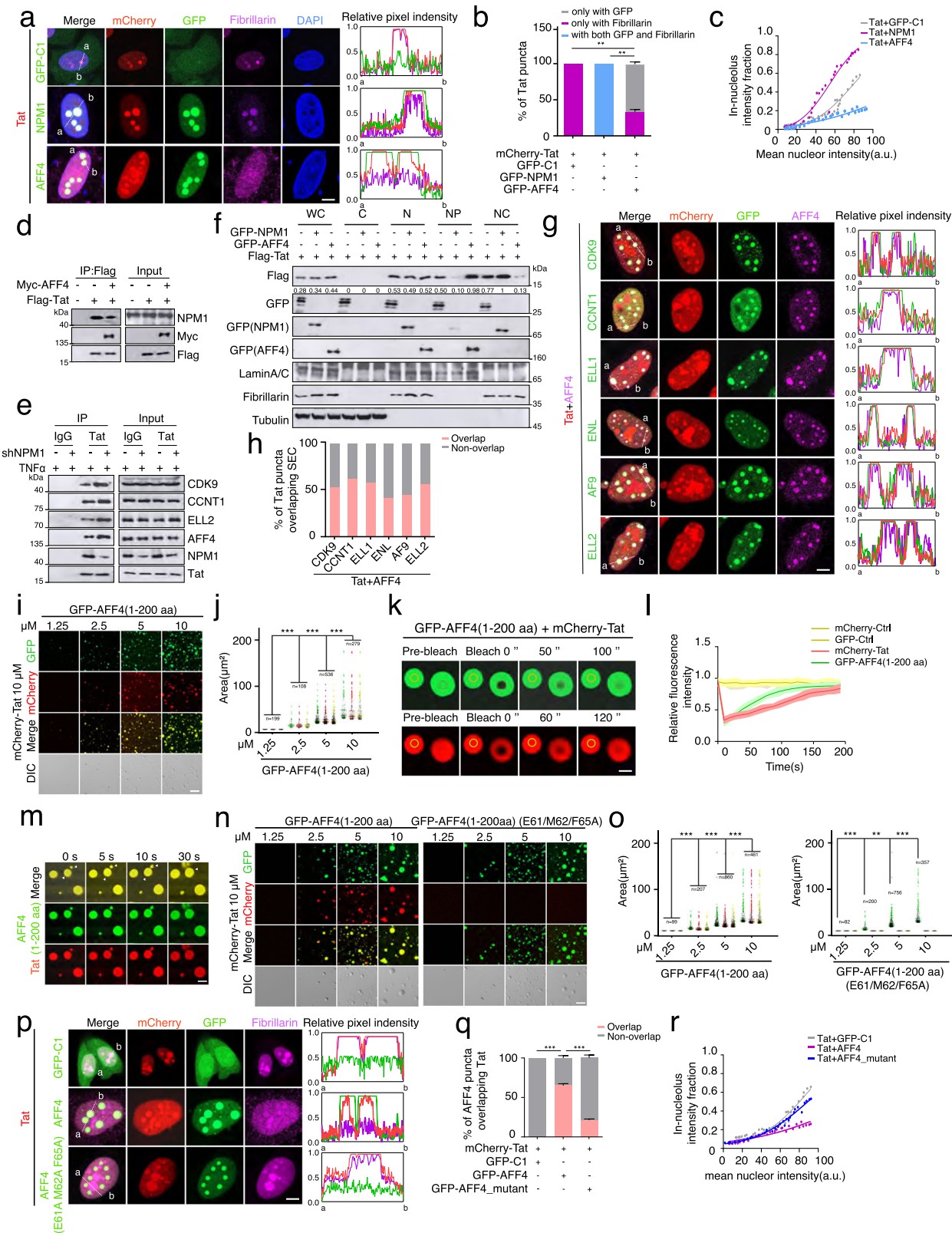

to incorporate the recombinant Tat into its droplets, even though it still underwent phase separation (Fig. 6n, o and Supplementary Fig. 6m). Similarly, the Tat mutant (K28A and F32A) that does not associate with AFF4, is also excluded from the AFF4 droplets in vitro (Supplementary Fig. 6n, o). Moreover, mutation of residues that mediate the interaction between Tat and AFF4 markedly diminishes

the nucleolar exit of Tat and its subsequent incorporation into the nucleoplasmic droplets induced by exogenous AFF4 (Fig. 6p–r and Supplementary Fig. 6p–r). Most importantly, the synergistic effects of AFF4 and Tat in transactivation of HIV-1 LTR were completely abolished when Tat phase separation into the SEC droplets was disrupted (Supplementary Fig. 6s). Together, these data demonstrated that the

**Fig. 6 | AFF4 phase separation incorporates Tat into the SEC droplets to activate HIV-1 transcription. a** Images (left) and line scan profiles (right) of Hela cells co-transfected with indicated constructs. **b** Bar plot showing percent Tat puncta co-localizing with AFF4 puncta (gray), fibrillarin puncta (purple), or both (cyan) in (**a**) (mean ± SD, $n = 3$). **c** Fraction of in-nucleolar Tat puncta fluorescence intensity as a function of mean nuclear intensity in (**a**). Each dot represents one cell ($n = 30$). **d, e** Nuclear extracts of 293T cells co-transfected with indicated constructs (**d**) or E4 cells stimulated with TNFα overnight (**e**) were subject to immunoprecipitation and immunoblotting analyses. **f** Immunoblotting analyses of the indicated subcellular fractions of Hela cells co-transfected with indicated constructs. WC whole cell, C cytoplasm, N nucleus, NP nucleoplasm, NC nucleoli. **g** Images (left) and line scan profiles (right) of Hela cells ectopically expressing the indicated protein. **h** Bar plots showing percent Tat puncta overlapping GFP dots in (**g**). **i** Images of in vitro droplets in solutions containing indicated recombinant proteins. **j** Quantification of the size of droplets in (**i**). **k** Photobleaching after recovery (FRAP) analyses of Tat-AFF4 heterotypic liquid droplets. **l** Quantification of FRAP signal in the bleach (red and green circle) and unbleached (yellow circle) areas in (**k**) (mean ± SD, $n = 7$). **m** Time-lapse confocal images of in vitro heterotypic Tat-AFF4 liquid droplets. **n** Images of liquid droplets in solutions containing indicated recombinant proteins. **o** Quantification of the droplet size in (**n**). **p** Images (left) and line scan profiles (right) of Hela cells ectopically expressing the indicated proteins. **q** Bar plot showing percent AFF4 puncta overlapping with Tat dots in (**p**) (mean ± SD, $n = 3$). **r** Fraction of in-nucleolar Tat puncta fluorescence intensity as a function of mean nuclear intensity in (**p**). Each dot represents one cell ($n = 30$). Scale bars in (**a, g, k, m**, and **p**, 5 μm) and in (**i** and **n**, 10 μm). Center line, median; box, upper and lower quantiles; whisker, maximal and minimal values (**j** and **o**). Statistical significance was determined using a two-tailed Student's $t$-test (**b** and **q**) and a two-sided Wilcoxon test (**j** and **o**). *$p < 0.05$; **$p < 0.01$; ***$p < 0.001$. Source data are provided as a Source Data file.

nucleolar exit of Tat and its subsequent incorporation into the nucleoplasmic SEC droplets are indispensable for the robust activation of HIV-1 transcription by Tat.

## PRMT2 promotes nucleolar sequestration of Tat and attenuates its incorporation into SEC droplets in a methylase-dependent fashion

Given that PRMT2 counteracts the formation of Tat-containing SEC in a methylase-dependent manner (Fig. 4), we investigated the impact of PRMT2 expression on AFF4-mediated Tat incorporation into SEC droplets. To this end, we co-expressed Tat along with either wild-type PRMT2 or methylase-defective PRMT2 in Hela cells and observed that cells expressing wild-type PRMT2 exhibited significantly higher levels of Tat accumulation in nucleoli compared to cells expressing GFP or methylase-deficient PRMT2 (Fig. 7a, b).

Subcellular fractionation experiments confirmed that the nuclear distribution of Tat is shifted to nucleoli in cells expressing wild-type PRMT2. In contrast, cells expressing methylate-deficient PRMT2 did not exhibit any notable effects on the nuclear distribution pattern of Tat (Fig. 7c). Conversely, depletion of PRMT2 markedly increased the levels of nucleoplasmic Tat along with notable concurrent reduction in the nucleolar Tat accumulation (Fig. 7d). These findings suggest that PRMT2 inhibits Tat-containing SEC formation by affecting Tat trafficking between nucleoli and nucleoplasm.

To assess the dynamic behavior of nuclear Tat, we performed a fluorescence recovery after photobleaching (FRAP) assay to compare the mobility of nucleolar Tat in cells expressing a vehicle, wild-type PRMT2 or a methylase-deficient variant. This assay revealed that expression of wild-type PRMT2 resulted in much slower fluorescence recovery kinetics of nucleolar Tat compared to cells expressing a vehicle or methylase-deficient mutant (Fig. 7e, f). At 18 min post photobleaching, vehicle and methylase-deficient PRMT2-expressing cells exhibited recovery of approximately 85% and 80% of initial fluorescent intensity, respectively, while wild-type PRMT2-expressing cells only reached about 50% of initial Tat fluorescent intensity (Fig. 7e, f). Additionally, depletion of PRMT2 markedly accelerated the fluorescence recovery of nucleolar Tat after photobleaching, as revealed by FRAP assay (Fig. 7g, h), indicating an enhanced mobility of nucleolar Tat in the absence of PRMT2. As nucleolar localization of Tat requires its association with NPM1[11], we reasoned that the altered mobility of nucleolar Tat upon PRMT2 expression changes may result from altered interaction between Tat and NPM1. As predicted, expression of a wild-type PRMT2 substantially increased the amount of NPM1 that co-precipitated with Tat, while the interaction between Tat and NPM1 was dramatically attenuated upon PRMT2 depletion (Supplementary Fig. 7a, b). These findings are consistent with the NPM1-induced reduction of Tat transactivation function in a luciferase reporter assay, spontaneous reactivation of proviral reactivation in E4 cells, as well as the diminished reestablishment of viral latency in 2D10 cells depleted for NPM1 (Supplementary Fig. 7c–e).

In accordance with the decreased association of Tat with SEC in the presence of PRMT2 (Fig. 4g, h), the expression of wild-type, but not a methylase-inactive PRMT2 markedly reduced the number of nucleoplasmic AFF4 droplets with Tat incorporation (Fig. 7i, j), further corroborating a crucial role of PRMT2 and its methylate activity in regulating Tat trafficking between nucleoli and nucleoplasm. To investigate whether the methylation of R52 and R53 by PRMT2 contributed to the nucleolar retention of Tat, we first examined the Tat methylation in distinct subcellular fractions and found that TatR52ame2 was specifically detected in nucleoli (Fig. 7k). Synergistic decreases in the interaction between Tat and NPM1, nucleolar accumulation of Tat in nucleoli, and much faster fluorescence recovery kinetics after photobleaching were observed upon mutation of R52, R53, or both to lysine (Fig. 7l–p and Supplementary Fig. 7f). Overall, these data confirmed that methylation of Tat R52 and R53 by PRMT2 is critically important for the nucleolar sequestration of Tat and the suppression of subsequent Tat incorporation into SEC droplets.

## Discussion
In this study, we conducted a targeted cDNA expression screening to identify potential PRMTs that have crucial regulatory roles in the Tat-mediated transactivation of HIV-1 transcription. Our results revealed that PRMT2, in addition to the known factors that we and others have previously reported, exerts a potent inhibitory effect on Tat activity in luciferase reporter assays. Further experiments using cell lines and primary CD4+ T cell models of HIV-1 latency, as well as HIV-1 infected CD4+ T cells from patients on ART confirmed that PRMT2 strongly suppresses proviral transcription, and its depletion by knocking down or CRSIPR-mediated knockout enhances spontaneous and LRAs-induced proviral reactivation. Moreover, PRMT2 ablation promotes productive infection and decelerates the latency re-entry of the reactivated provirus. Tat is known to reside in both nucleoplasm and nucleoli and activates HIV-1 transcription by triggering SEC-dependent release of paused RNA Pol II from the HIV-1 promoter into the processive elongation stage. Our study revealed that the competitive association of Tat with NPM1 and AFF4 determines its nuclear distribution. Specifically, NPM1 interacts with Tat and recruits it to the nucleoli, while binding to AFF4 leads to its incorporation into the nucleoplasmic SEC liquid droplets through AFF4 phase separation. We also found that PRMT2 preferentially methylates Tat at R52 to favor its association with NPM1 and nucleolar sequestration, thereby attenuating its incorporation into the nucleoplasmic SEC droplet to suppress HIV-1 transcription and promote viral latency (Supplementary Fig. 7g).

Tat is an essential viral transactivator for both HIV-1 transcription and proviral reactivation from latency[24,38]. Variations in the Tat protein sequence are associated with the differential replicative capacity of HIV-1 strains[13]. To prevent cytopathic effects of uncontrolled viral replication and to establish long-term latent infection in host cells, the transcriptional activity of Tat must be strictly controlled to alternate

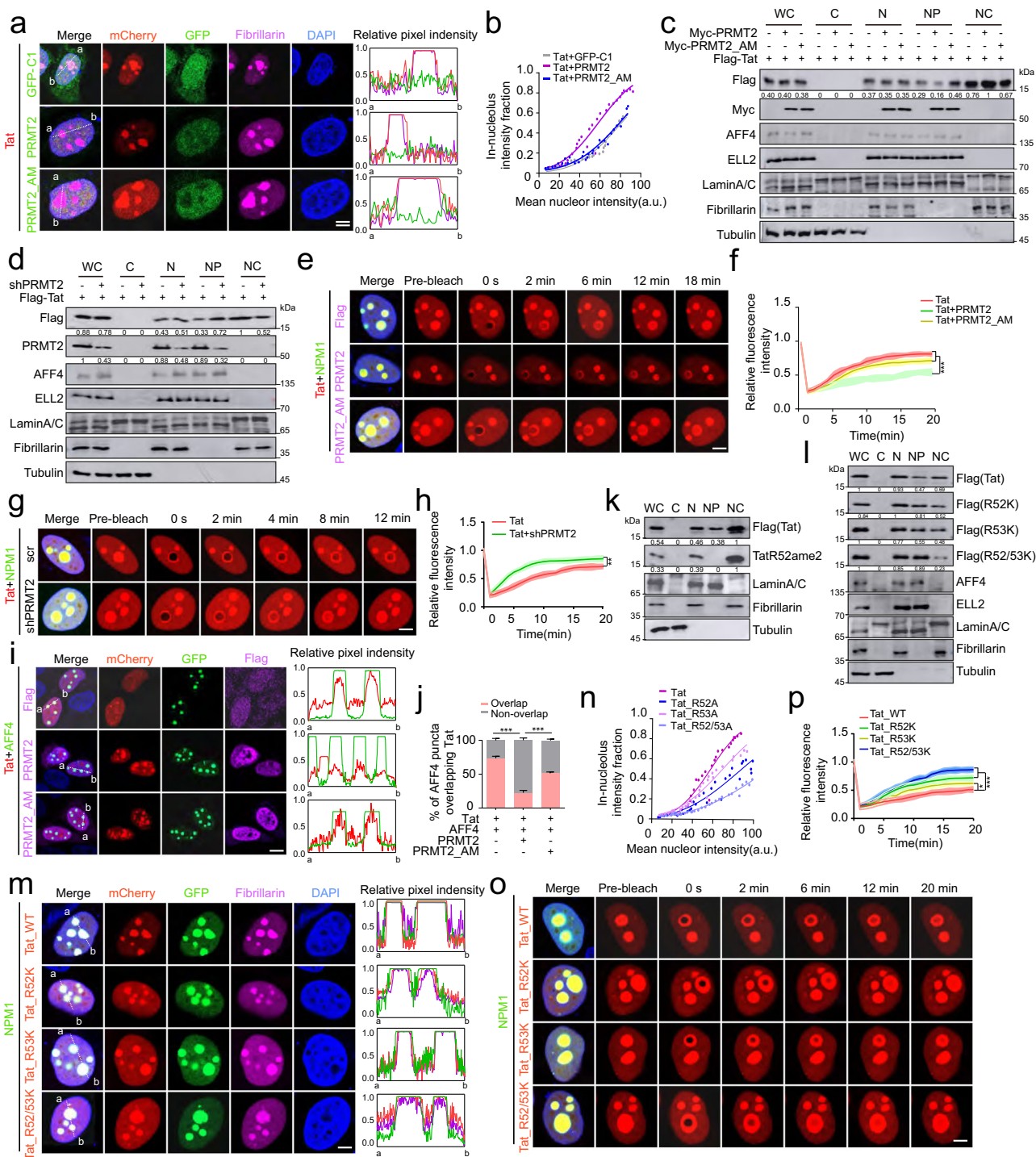

between active and latent states. Post-translational modifications, such as acetylation, phosphorylation, methylation, and ubiquitination, can modulate Tat's transcriptional activity by affecting interactions between Tat, P-TEFb, and TAR[13,39]. Furthermore, Tat traffics throughout infected cells, and its transcriptional activity is closely linked to its subcellular localization. For example, nucleolar sequestration of Tat by a nucleolar TAR decoy inhibitor or forced translocation to the cytoplasm by an engineered binding partner leads to a considerable reduction in Tat activity[11,12]. However, the extent to which HIV-1 exploits nucleolar sequestration of Tat to regulate its transcription and infection status in infected cells, and the underlying regulatory mechanisms remain to be determined. Our study revealed that PRMT2 suppresses Tat transcriptional activity and promotes the

establishment and maintenance of proviral latency through facilitating NPM1-mediated Tat nucleolar sequestration, proving that nucleolar translocation of Tat is a potential critical mechanism utilized by HIV-1 to achieve and sustain latent infection.

The nucleolar localization of Tat requires the interaction of its highly basic ARM region with the NoLS-binding domain of NPM1[11]. Our biochemical analysis demonstrated that PRMT2 preferentially methylates the R52 within the ARM region of Tat, which promotes its nucleolar localization by enhancing its association with NPM1. However, the mechanism by which R52 methylation by PRMT2 increases Tat binding to NPM1 remains unclear at this point. As the NoLS-binding domain of NPM1 is also basic, it has been proposed that the NPM1-Tat interaction is mediated by the spatial conformation rather than the

**Fig. 7 | PRMT2 attenuates Tat incorporation into nucleoplasmic AFF4 droplets by promoting arginine methylation-dependent nucleolar compartmentalization of Tat. a** Images (left) and line scan profiles (right) of Hela cells co-expressing indicated proteins. **b** Fraction of in-nucleolar Tat puncta fluorescence intensity as a function of mean nuclear intensity in (**a**). Each dot represents one cell ($n = 30$). **c**, **d** Immunoblotting analyses of different subcellular fractions of Hela cells ectopically expressing flag-Tat alone or together with wild-type or methylase-deficient mutant PRMT2 (**c**), or with control or PRMT2-specific shRNA (**d**). **e** Time-lapse confocal images of nucleolar Tat before and after photobleaching in HeLa cells ectopically expressing indicated proteins. **f** Quantification of FRAP signal over time for the nucleolar Tat in (**e**). **g** Time-lapse confocal images of nucleolar Tat before and after photobleaching in HeLa cells infected with indicated shRNA. **h** Quantification of FRAP signal over time for the nucleolar Tat in (**g**). **i** Images (left) and line scan profiles (right) of heterotypic Tat-AFF4 puncta in HeLa cells co-expressing indicated proteins (left). **j** Bar plot showing the proportion of AFF4 droplets overlapping with Tat puncta in (**i**) (mean ± SD, $n = 3$). **k** Immunoblotting analyses of methylated and unmodified Tat in different subcellular compartments of Hela cells transfected with a flag-tagged Tat. **l** Immunoblotting analyses of different subcellular compartments of Hela cells transfected with a flag-tagged wild-type and indicated mutant Tat. **m** Images (left) and line scan profiles (right) of nucleolar Tat puncta in Hela cells co-expressing indicated proteins. **n** Fraction of in-nucleolar Tat puncta fluorescence intensity as a function of mean nuclear intensity in Hela cells in (**m**). Each dot represents one cell ($n = 30$). **o** Time-lapse confocal images of nucleolar Tat before and after photobleaching in HeLa cells co-expressing indicated proteins. **p** Quantification of FRAP signal over time for the nucleolar Tat in (**o**). Quantifications in (**f**, **h** and **p**, $n = 7$ droplets) are shown as mean ± SD. Scale bars in (**a**, **e**, **g**, **m** and **o**, 5 μm) and in (**i**, 10 μm). Values from densitometric analyses of immunoblots are shown beneath respective gel bands. Statistical significance was determined using a two-tailed Student's $t$-test (**j**) and two-way ANOVA test (**f**, **h** and **p**). *$p < 0.05$; **$p < 0.01$; ***$p < 0.001$.

electrostatic interactions[11]. Thus, we hypothesized that R52 methylation might modulate the conformation of Tat to facilitate access to the NoLS-binding domain of NPM1. Future structural studies are warranted to confirm this assumption. Besides PRMT2, R52 and R53 were previously shown to be methylated by PRMT6, which inhibits Tat transcriptional activity and viral replication by disrupting the formation of the Tat-TAR-P-TEFb quaternary complex[14,15]. Whether PRMT2 and PRMT6 cooperate with each other or function separately at different stages of viral infection to promote transcriptional silencing and viral latency remains to be elucidated.

It has been well documented that Tat activates HIV-1 transcription and replication primarily through recruiting the SEC to the viral promoter to trigger the release of paused Pol II into productive elongation[3]. However, the molecular mechanisms for how Tat-mediated SEC recruitment leads to highly efficient transcriptional elongation of viral gene remain incompletely understood. Recent studies have demonstrated that several components of the SEC can undergo phase separation into liquid droplets to mediate hyperphosphorylation and efficient elongation of Pol II[6–8]. Due to Tat's strong physical association with multiple subunits of the SEC, it is presumable that Tat may exploit the phase separation properties of the SEC to achieve robust activation of HIV-1 transcription. Indeed, our data provided compelling evidence to support this notion and demonstrated that Tat is readily incorporated into the nucleoplasmic SEC droplets via the phase separation of AFF4, but not other components though some of them also undergo phase separation themselves. AFF4 phase separation-mediated Tat recruitment into the SEC droplets relies on the direct interaction between AFF4 and Tat, which is attenuated by PRMT2 in a methylase-dependent manner. Our findings suggest that PRMT2-induced dissociation of Tat from the SEC droplets may act as a previously unrecognized mechanism that shuts down HIV-1 transcription and promotes entrance into proviral latency.

While LRAs with distinct mechanisms of action have demonstrated efficacy in reactivating proviral latency in vitro within latently infected cell lines, their performance in reducing the size of latent reservoirs in patients has been limited. For instance, the effectiveness of JQ1 in reversing HIV-1 latency exhibits significant variability when applied to ex vivo CD4+ T cells isolated from individuals undergoing antiretroviral therapy (ART) with HIV-1 infections[40–42]. Notably, a substantial induction of HIV-1 transcripts has been limited to a subset of patients' CD4+ T cells following a 24-h ex vivo treatment[40,43]. This divergent reactivation of HIV-1 latency within patient cells can likely be attributed to the complex and heterogeneous nature of the latent reservoir of HIV-1 within CD4+ T cells, which demonstrates substantially variable responses to proviral reactivation induced by different families of LRAs[43,44]. Consistent with previous research[21,43], our studies have revealed a significant and consistent reactivation of latent provirus in CD4+ T cells of four HIV-1 infected individuals on ART after extended exposure to JQ1. This indicates that latent proviruses within the heterogeneous reservoirs exhibit different reactivation kinetics in response to LRAs, and more latent proviruses can be reactivated after prolonged exposure to JQ1.

Our work has demonstrated that PRMT2 suppresses HIV-1 transcription and sustains proviral latency in a methylase-dependent manner. Depletion of PRMT2 can enhance the efficacy of a select few LRAs, including JQ1, in reactivating proviral transcription in patient-derived CD4+ T cells. This underscores the potential value of incorporating PRMT2 inhibitors alongside existing LRAs to target latent proviruses across a broader spectrum of heterogeneous reservoirs. Presently, there are no selective inhibitors available for PRMT2. However, it's worth noting that the atomic structure of evolutionarily conserved zebrafish and mouse PRMT2 has been elucidated[17,45]. This structural information holds the promise of expediting the design and development of selective PRMT2 inhibitors with the potential to reactivate latent proviruses and eliminate heterogeneous reservoirs in conjunction with currently available LRAs

In summary, our work has identified PRMT2 as a novel host restriction factor that potently suppresses Tat transcriptional activity and promotes proviral latency through preferential methylation of Tat at R52. The methylation of Tat by PRMT2 enhances its binding to NPM1 and causes its sequestration in the nucleoli, which reduces the availability of nucleoplasmic Tat for recruitment into the SEC droplets by AFF4 phase separation to activate HIV-1 transcription. The pharmacological inhibition of PRMT2, which blocks nucleolar translocation and dissociation of Tat from the SEC droplets may have the potential to synergize with the existing LRAs to reactivate latent provirus and eliminate latent viral reservoirs.

## Methods
### Cell culture and reactivation
HEK293T and Hela cells were obtained from the American Type Culture Collection and maintained in Dulbecco's modified Eagle's medium (DMEM; Biological Industries, 06-1055-57-1ACS) and 10% fetal bovine serum (FBS; Biological Industries, 04-001-1ACS). Jurkat T cells and their E4 and 2D10 clonal cell lines latently infected with pseudotyped HIV-1 reporter provirus were cultured in RPMI 1640 (Biological Industries, 01-100-1ACS) supplemented with 10% FBS (Biological Industries, 04-001-1ACS), 2.0 mM L-glutamine, and 1% penicillin/ streptomycin.

For reactivation of HIV-1 provirus, E4 and 2D10 cells were stimulated with TNFα (10 ng/ml), PMA (50 ng/ml) and JQ1 (1 μM) for indicated times and then proceeded to downstream analyses specified in the relevant figure legends. For kinetic analyses of the reversion of reactivated provirus to latency, 2D10 cells were treated with 10 ng/ml overnight to achieve full activation of proviral transcription. Cells were then washed twice with phosphate-buffered saline (PBS) and resuspended in complete growth media followed by analyses at the indicated time points following TNFα withdrawal. (25 μl/ml) CD3/CD28 antibodies and (50 ng/ml) PMA were used to activate normal CD4+

T cells for indicated times. Cell lines were authenticated genetically by the providers and we indirectly verified their identity by their morphology, growth behavior or transcriptomic profiles. All cell lines were regularly tested for mycoplasma contamination using the MycoBlue Mycoplasma Detector (Vazyme, D101-01) and maintained at 5% $CO_2$ and 37 °C.

### Primary CD4[+] T cell isolation

Peripheral blood mononuclear cells (PBMCs) were isolated from whole blood using a density gradient centrifugation over the Ficoll medium (TBDScience, LTS1077). Resting primary CD4[+] T cells were purified from PBMCs using negative selection by the EasySep Kit (STEMCELL Technologies, 17952) according to the manufacturer's instructions. HIV-1 infected CD4[+] T cells were purified the same as the primary CD4[+] T cells from healthy donors. Both primary CD4[+] T cells were cultured in RPMI 1640 (Biological Industries, 01-100-1ACS) supplemented with 10% FBS (HyClone, SH30406.05), 2.0 mM L-glutamine, and 1% penicillin/ streptomycin and IL-2 (100 U/ml, Sino Biological, GMP-11848-HNAE).

### Lentivirus production and infection

For the preparation of pLKO and pSin-based lentivirus, corresponding constructs were co-transfected into HEK293T cells with the psPAX2 and pMD2.G plasmids at a ratio of 2:2:1 by Lipofectamine 2000 (Thermo Fisher Scientific, 11668027) following the manufacturer's instructions. Pseudotyped HIV-1$_{GKO}$ and HIV-1$_{NL-3/Luciferase}$ virus were generated by co-transfecting HIV-1$_{GKO}$ or HIV-1$_{NL-3/Luciferase}$ plasmid with pCMV-VSV-G vector into HEK293T cells at a ratio of 2:1 using Lipofectamine 2000. The viral particles were harvested at 48 and 96 h after transfection, filtered by 0.45 μM filter unit (Millipore), concentrated by ultracentrifugation at 90,000 × g, 4 °C for 90 min, 4 °C, and then stored at −80 °C in aliquots.

For cell line infection, cells were transduced with lentivirus in the presence of polybrene (4 μg/ml; Sigma-Aldrich) for 24 h and selected with puromycin (2 μg/ml) for 48 h to eliminate non-transduced cells before collection for analysis at indicated time points. For primary CD4[+] T infection, cells were stimulated with anti-CD3/CD28 (25 μl/ml) for 24 h and then first spinoculated with HIV-1$_{GKO}$ viruses at 800 × g, 30 °C for 90 min followed by an additional 10 h incubation before withdrawal of viruses. Cells were harvested for analyses 72 h post-infection.

### HIV reactivation in primary CD4[+] T cells of healthy donors and HIV-1 infected patients

Isolated CD4[+] T cells were seeded at a density of $1 \times 10^6$ per well in 150 μl of complete growth medium in a 96-well flat-bottom plate and activated with ImmunoCult Human CD3/CD28 T Cell Activator (STEMCELL Technologies, 10791) and 200 IU IL-2 for 48 h followed by transduction with pseudotyped HIV-1$_{NL-3/Luciferase}$ viruses. At day 10, cells were infected by PRMT2 or PRMT2_A/M-expressing virus and after 10 h infected and uninfected CD4[+] T cells were isolated by 2 μg/ml puromycin. At day 12, latently infected CD4[+] T cells were activated with PMA (10 ng/ml) or ImmunoCult Human CD3/CD28 T Cell Activator (STEMCELL, 10970) for 24 h. Reactivation of HIV provirus was determined by examining the luciferase activity.

For latency reversal in primary CD4[+] T cells of HIV-1 infected individuals, purified CD4[+] T cells were infected with control and PRM2-targeting shRNAs in the presence of polybrene. Forty-eight hours post-infection, cells were stimulated with DMSO, JQ1(0.5 and 1 μM), PMA(25 and 50 ng/ml), TSA (25 and 50 nM), and anti-CD3/CD28 for an additional 72 h before harvesting for total RNA extraction.

### Luciferase reporter assay

For luciferase reporter assays, HEK293T cells were co-transfected with a reporter vector expressing firefly luciferase under the control of HIV-1 LTR, LacZ internal control plasmid, and constructs encoding the indicated protein or shRNAs by Lipofectamine 2000. The total amount of transfected plasmids per well of transfection was adjusted to be equal. Six hours after transfection, cells were treated with MS023 (1 μM and 2 μM), HLCL-61 HCL (8.5 μM and 17 μM), flavopiridol (20 nM), JQ-1(1 μM), PMA (50 ng/ml) or KL-2 (10 μM). Thirty-six hours after transfection, cells were harvested and lysed for quantitation of firefly luciferase and LacZ activities on the Synergy HT platform (BioTek). Luciferase activity was normalized to the β-galactosidase internal control signal and compared to that in the control cells, which was arbitrarily set to 1.

### Flow cytometry

Cells were washed twice with ice-cold PBS, pelleted by centrifugation, resuspended, and then incubated on ice for 10 min in pre-cooled fluorescence-activated cell sorting (FACS) buffer (PBS and 3% FBS) supplemented with 4′,6-diamidino-2-phenylindole (0.1 g/ml). d2EGFP fluorescent signal in E4 or 2D10 cells with or without treatment were measured on FACSCalibur flow cytometer (BD Biosciences) using the BD FACSDivaTm software and analyzed by FlowJo X software.

### Quantitative reverse transcription polymerase chain reaction

Total RNA was isolated using TRIzol, treated with deoxyribonuclease I (New England Biolabs, M0303S), and reverse transcribed using SuperScript III and random primers (Invitrogen). The resulting cDNA was analyzed by quantitative PCR using Green SuperMix for iQ (VVR, 01414-144) on a 7500 Fast Real-Time PCR system (Thermo Fisher Scientific) or on a CFX ConnectTM Real-Time PCR Detection System (Bio-Rad).

### Flag purification and mass spectrometric analyses

Flag-Tat was transfected into HEK293T cells, and the nuclear extract was prepared according to the Dignam method. Tat-bound proteins were purified on anti-Flag (M2) agarose beads in the presence of Benzonase (Novagen, 70664), separated on SDS–polyacrylamide gel electrophoresis gel, and visualized by silver staining. Trichloroacetic acid–precipitated protein mixtures from the Flag purifications were digested with endoproteinase Lys-C and trypsin (Roche, 11058533103) and analyzed by MudPIT.

### Glutathione S-transferase pull-down

Glutathione S-transferase (GST), GST fusion proteins were expressed in Rosetta bacteria and immobilized on glutathione sepharose 4B beads (Sigma, G0924), washed three times with PBS, and then resuspended in RIPA buffer supplemented with 1 mM $MgCl_2$ and proteinase inhibitors (Sigma-Aldrich). GST, GST fusion protein-bound beads (1 ml) were incubated for 3 h at 4 °C with His-tagged proteins, which were expressed in Rosetta and purified with Ni-NTA resins (QIAGEN, 30310) according to the manufacturer's instructions. Beads were washed with RIPA buffer three times, and the bound proteins were eluted in SDS loading buffer, followed by Coomassie Bue staining or by western blotting with indicated antibodies.

### Immunoprecipitation

Cells were washed with ice-cold PBS twice, resuspended in Dignam hypotonic buffer [10 mM Hepes (pH 7.9), 1.5 mM $MgCl_2$, and 10 mM KCl], incubated on ice for 10 min, followed by brief centrifugation for nuclei precipitation. The nuclei were resuspended in radio immuno-precipitation assay (RIPA) buffer [20 mM tris-HCl (pH 7.4), 150 mM NaCl, 1% NP-40, 1% sodium deoxycholate, 0.1% SDS, and 1 mM dithiothreitol], supplemented with 1 mM $MgCl_2$ and proteinase inhibitors (Sigma-Aldrich), and lysed with gentle agitation for 30 min at 4 °C. After centrifugation at 13,000 rpm for 30 min, the supernatant was incubated with the Flag agarose (Sigma, A2220) or indicated antibodies and Protein A/G PLUS agarose (Santa Cruz Biotechnology, sc-2003) in the

presence of Benzonase (1 U/ml; Novagen, 70664) at 4 °C overnight with gentle rotation. The beads were spun down and washed three times with a wash buffer [10 mM Hepes (pH 7.4), 1 mM MgCl$_2$, 300 mM NaCl, 10 mM KCl, and 0.2% Triton X-100] before boiling in SDS loading buffer.

## Chromatin immunoprecipitation

For this, $1 \times 10^7$ to $3 \times 10^7$ treated or untreated cells were cross-linked with 1% paraformaldehyde at room temperature for 10 min and quenched by glycine. Cells were sonicated or generate chromatin fragments of 200 to 600 bp with a Sonics Vibra-Cell sonicator (Sonics & Materials) followed by immunoprecipitation with the indicated antibodies. The resulting DNA was analyzed using SYBR Green Mix on the CFX connect Real-Time PCR detection System (Bio-Rad) using Bio-Rad CFX Maestro software and normalized to input.

## Glycerol gradient ultracentrifugation

Glycerol gradients (8% to 40%) were prepared freshly with MRIPA buffer containing 25 mM HEPES (pH 7.9), 150 mM NaCl, 1% NP40, 1 µg/µl each of protease inhibitors aprotinin, leupeptin, and pepstatin and glycerol by Automatic Density Gradient (Biocomp). Then, 500 µg of cleared nuclear extract (brought to 200 µl with MRIPA buffer) was slowly layered on top of a 13 ml glycerol gradient and ultracentrifuged in a swinging-bucket rotor (Beckman, Optimal-100XP,SW 41 Ti) at 34,000 rpm for 16 h (accel = 8, decel = 8). After centrifugation, 200 µl per fraction was manually collected from the top of the gradient tube and analyzed by immunoblotting with indicated antibodies.

## CRISPR/Cas9-guided knockout

sgRNA oligos targeting PRM2/PRMT6 were cloned into the lenti-CRISPR V2 vector (Addgene, #52961) and the resulting constructs were transfected into HEK293T cells followed by selection with 2 µg/ml puromycin for 48 h. The polyclonal cells were analyzed by immuno-blotting for the expression of PRTM2 and PRMT6 and transduced with indicated lentiviral expression constructs for rescue experiments.

## Methylation assays

Recombinant his-tagged wild-type Tat or its mutant (2 µg) was incubated at 30 °C for 4 h with purified flag-tagged wild-type PRMT2 or methylase-inactive mutant in a buffer containing 25 mM Tris-HCl (pH 7.5), 5 mM MgCl$_2$, 1 mM DTT the presence of S-Adenosyl methionine (SAM,APE×BIO,B3513) or 0.55 µCi of [methyl-3H]S-adenosyl-l-methionine ($^3$H-SAM, NET155V250UC, 1 mCi/Ml, PerkinElmer) and in a final volume of 20 µl. Reactions were stopped by adding 5 µl of 5× Laemmli buffer, followed by heating at 100 °C for 5 min. Samples were resolved on 12 % SDS–PAGE gel, and visualized by silver staining or analyzed by immunoblotting with indicated antibodies. Reactions performed with radioactive SAM, protein bands of Tat on SDS-PAGE gel were excised based on their molecular size and subject to liquid scintillation analyses.

## Dot blots assay

Polyclonal antibodies against asymmetrically dimethylated R52 of Tat (Tat_R52ame2) were generated by immunizing two rabbits with KLH conjugated synthetic peptide (YGRKK-Rame2-RQRRR-C-KLH) in a 70-day rabbit peptide protocol (ABclonal). To evaluate the specificity of Tat_R52ame2 antibodies, synthetic methylated and corresponding unmodified peptides were spotted onto a nitrocellulose membrane. Membrane was dried at room temperature and blocked in PBST buffer (20 mm Tris-HCl, pH 7.5, 150 mm NaCl, 0.05% Tween 20) with 5% non-fat milk. After washing, the membrane was probed with Tat R52ame2 antibodies.

## Subcellular fractionation

For preparation of whole cellular extract (WC), Hela cells with or without infection of lentivirus encoding indicated protein were washed twice with ice-cold PBS and lysed in RIPA buffer (50 mM Tris-HCl, pH 8.0, 150 mM NaCl, 0.5% sodium deoxycholate, 1% NP-40, and 0.1% SDS) for 20 min on ice. Cell lysates were sonicated in ice with 30 pulses of 1 s on and 1 s off at 30% amplitude and then centrifuged to collect supernatant, which is whole cellular extract. For separation of cellular extract into cytosolic, nuclear, nucleoplasmic and nucleolar fractions, $5 \times 10^6$ cells were resuspended in hypotonic buffer (20 mM Tris-HCl, pH 7.9, 1.5 mM MgCl$_2$, 10 mM KCl and 1 mM DTT) with a final concentration of 0.5% NP-40 and incubated on ice for 20 min. After centrifugation at $300 \times g$, 4 °C for 15 min, the supernatant was retained as cytosolic fraction (C) and the pellet (crude nuclei) was resuspended in RPIA buffer for preparation of nuclear fraction (N).

For preparation of nucleoplasmic and nucleolar extracts, crude nuclei were purified by resuspending in solution 1 (0.25 M sucrose, 10 mM MgCl$_2$), layered over a cushion of solution 2 (0.35 M sucrose, 0.5 mM MgCl$_2$), and centrifuged at $1500 \times g$ for 15 min. The pellet (clean nuclei) was resuspended in solution 2 and sonicated in ice with 6 pulse of 10 s on and 10 s off at 30% amplitude. The sonicated sample was layered over solution 3 (0.88 M sucrose, 0.5 mM MgCl$_2$) and centrifuged at $2800 \times g$, 4 °C for 15 min. The supernatant was reserved as nucleoplasmic extracts (NP) and the pellet (nucleoli) was resuspended in RPIA buffer, sonicated in ice with 30 pulses of 1 s on and 1 s off at 30% amplitude and centrifuged to collect supernatant (nucleolar fractions, NC).

## Immunofluorescence staining and confocal microscopy

Hela cells grown on cover glasses in a 6-well plate were transfected with EGFP- or mCherry-based expression vectors. Thirty-six hours post-transfection, cells were washed twice with PBS, fixed in 4% paraformaldehyde and then permeabilized with PBS containing 0.25% Triton X-100 for 10 min at room temperature. After that, cells were blocked with 1% BSA diluted in PBS for 1 h to block the nonspecific binding of the antibody. Cells were incubated with primary antibody at 4 °C overnight. After incubation, cells were washed with PBST three times and then incubated with fluorophore-conjugated secondary antibody at room temperature for 1 h in the dark. After three washes, cells were counterstained with 250 ng/ml of DAPI for 10 min to label the nucleus and were mounted with anti-fade-mounting medium. Images were visualized and captured on an inverted laser scanning confocal microscope (Zeiss, Axio-Imager_LSM-800) with 63×oil DIC objective under the control of the Zeiss Zen software.

## In vitro droplet formation and quantification

His-tagged recombinant proteins used in vitro droplet formation assay were expressed in Rosetta and purified with Ni-NTA resin according to the manufacturer's instructions. After elution, proteins were dialyzed overnight at 4 °C against 150 mM NaCl, 20 mM Tris pH 7.5, and 1 mM DTT and concentrated with Amicon ultra centrifugal filters (Millipore) and stored at −80 °C in aliquots. The purity of the proteins was examined by SDS-PAGE and Coomassie Blue staining, and concentrations were determined using known concentrations of bovine serum albumin (BSA) as standards.

Recombinant proteins were added to a final concentration as indicated in the relevant figures or their legends in a buffer containing 20 mM Tris-HCl pH 7.5, 1 mM DTT, 150 mM NaCl and 10% PEG8000. The protein solution was trapped between two coverslips immediately after dilution and imaged on a Zeiss microscope (Axio-Imager_LSM-800) with a 20× objective. The number and size of droplets were analyzed using FIJI. All images were equally thresholded (>3 pixel minimum droplet size), and droplet area and number were identified using the Analyze Particles function of FIJI.

## FRAP assay and data analysis

For this, $1 \times 10^4$ cells were seeded on glass-bottom cell culture dishes and transfected with indicated plasmids. At 36 to 48 h after

transfection, images were captured at room temperature on an inverted laser scanning confocal microscope (Zeiss, Axio-Imager_LSM-800) with 63×oil DIC objective under the control of the Zeiss software. Photobleaching of EGFP-tagged and mCherry-tagged proteins were accomplished using 15% laser power (561 nm laser line) and 30% laser power (488 nm laser line) for 2 s, respectively. Images were collected before bleaching and every 10 and 30 s post bleaching of in vitro droplets and nucleolar Tat protein.

Postbleach FRAP recovery data of every indicated time representing the mean fluorescence intensity of the monitored regions were background subtracted and normalized to prebleach intensity. Postbleach FRAP recovery data were averaged over seven biological replicates for each experiment.

## Ethics
Whole blood of anonymous health donors was purchased from Tianjin Blood Center. No informed consent was needed as we did not have any interaction with these healthy donors or protected information. Whole blood of chronically HIV-1-infected individuals on ARTs was collected at Tianjin Second People's Hospital, China, which is a designated hospital for the treatment of HIV-1-infected patients. Participants were selected based on their well-documented persistent viral suppression with undetectable plasma viremia (<50 copies/ml) and high CD4$^+$ T cell number (>500 cells/ml) in peripheral blood for at least two years. All patients provided written informed consent and the study was reviewed and approved by the Ethics Committees of Tianjin Second People's Hospital (Reference number, LL-BG-032). The use of patients' samples is in accordance with the Declaration of Helsinki.

## Statistics and reproducibility
Sample size, P-values and error bars are described in the text or figure legends. Statistical significance was determined using GraphPad software v.9.4 with the tests specified in the figure legends. No statistical method was used to predetermine sample size and no data were excluded from the analyses. All experiments were repeated at least twice with similar results obtained.

## Reporting summary
Further information on research design is available in the Nature Portfolio Reporting Summary linked to this article.

## Data availability
All information about reagents, primers, antibodies, generated plasmids, etc. is provided in the source data file. GuavaH database[28] is accessible via http://www.GuavaH.org. Source data are provided with this paper.

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

## Acknowledgements

We thank the Core Facility of Research Center of Basic Medical Sciences at Tianjin Medical University for technical assistance. These studies were supported by starting funds of Tianjin Medical University, the National Natural Science Foundation of China (32270650, 31872825, 32022013), and Tianjin Natural Science Foundation (18JCYBJC42400, 20JCJQJC00290) to D.H., National Natural Science Foundation of China (32000469) and Tianjin Municipal Science and Technology Commission Grant (21JCYBJC01230) to X.G., and the Health Science and Technology Project of Tianjin Health Commission (ZC20037) to P.M.

## Author contributions

D.H. and X.G. conceived the project and supervised the study. J.J. and H.B. performed the majority of the biochemical, molecular and cellular experiments. H.N. optimized condition for virus infection of normal CD4+ T cells. Y.H. performed all flow cytometry analyses. T.L., R.X. and X.B. conducted mass spectrometry analyses under the supervision of K.Z. and K.L. L.F. and P.M. collected peripheral blood and guided the handling and operation of HIV-1-infected patient cells. T.D., Z.L. and X.W. provided helpful discussion throughout the project. D.H., X.G. and J.J. drafted the manuscript with the input from all authors. All authors contributed to editing the manuscript.

## Competing interests

The authors declare no competing interests.
