## [Peer Review File · Nature Communications]

Reviewers' Comments:

Reviewer #1:

Remarks to the Author:

This work innovatively identified that PRMT2 plays a role in promoting HIV latency. It further elucidated a potentially new mechanism that PRMT2 interacts with HIV Tat protein and leads to its methylation at R52, which attenuates Tat incorporation into the nucleoplasmic AFF4 droplets, thus reducing Tat-SEC complex and preventing Tat-mediated HIV proviral transactivation. Overall, the results are convincing, which confirms PRMT2 as a novel host epigenetic factor that silences HIV proviruses. However, there are a few major issues to solve before the consideration for publication.

- 1). Most mechanistic studies were performed only in cell lines. It needs to show that PRMT2 indeed interacts with Tat protein and methylates Tat at R52 in primary CD4+ T cells. Likewise, the impact of PRMT2 on Tat-SEC/AFF4 protein complex also needs to be determined in primary CD4+ T cells.
- 2). There was no result to show that CRISPR causes the complete knockout of endogenous PRMT2.
- 3). Certain key protein immunoblotting data were quite subtle and not convincing. Protein bands need to be quantified. Statistic analysis shall be further performed from multiple independent protein immunoblotting experiments.
- 4). Earlier studies showed that JQ1 fails to reactivate latent HIV in patients' PBMCs. However, in Fig 3J, JQ1 was effective to reactivate latent HIV in primary CD4+ T cells isolated from HIV+ subjects. The authors shall discuss such discrepancy.
- 5). It seems that no PRMT2-specific inhibitors are currently available. Thus, its therapeutic potential for HIV lytic reactivation and "shock and kill" cure is not clear. The authors shall discuss this issue.

Reviewer #2:

Remarks to the Author:

The paper by Jin et al focuses on the role of PRMT2, a methyltransferase, and its potential to inhibit Tat mediated HIV transcription which impacts establishment and maintenance of latency. By knocking down and overexpressing PRMT2 in variety of cell lines and primary cells they provide biochemical and functional evidence that PRMT2 influences Tat function. They biochemically map key domains and key interacting factors which include components of the Super Elongation Complex. Finally, they make an argument that a role of PRMT2 is to influence nuclear distribution of these factors. Strengths of the paper are the exhaustive mapping of biochemical activities and interactions of PRMT2 with Tat and the SEC, the use of Tat and PRMT2 mutations to validate knock-down and overexpression experiments, and the use of primary cells, including clinical samples that support the argument for PRMT2 influencing latency. Modest weaknesses include that there are limited data, examining the role of endogenous PRMT on HIV infected cells, there may be a missed opportunity to explore potential functional interactions between PRMT2 and PRMT6, a previous methyltransferase described to regulate HIV transcription, and, although nuclear distribution of Tat and AFF4 were convincing, the PRMT2 role was less so, making this feel tangential to the overall PRMT2 story. Some specific comments are below.

1. All experiments were performed with cells that had factors overexpressed or knocked down. Although these experiments are essential for getting to mechanism and are convincing, it would validate these results if experiments were performed demonstrating Tat methylation, co-IPs, ChIPs, etc, using infected cells that expressed normal endogenous levels of PRMT2.
2. Expression of PRMT6 should be examined as a control to assure that PRMT2 is not influencing the expression of this PRMT family member. Related to this, examining the physical and functional interactions of PRMT2 and PRMT6 in HIV infected cells seems like an important question that could have been addressed.
3. In fig 2 the reversal of latency takes several days. This seems somewhat surprising in the context of transcriptional regulation. Is there a selection of cells or some change in cell phenotypes observed with the diminished PRMT2 expression? Does this provide any insights into PRMT2 function? Also, figure 2 I and J were unclear as to what was 100% since they are observing maybe

only 20% infected cells; how were these profiles generated?

4. Experiments show changes in Tat-mediated HIV-1 transcription but Tat still induces some transcription. Were experiments performed to provide insights into the impact PRMT2 has on HIV-1 production.

5. The hypothesis that PRMT2 influences Tat and AFF4 phase separation are intriguing. However, the blots in Fig 7 for PRMT2 suggested subtle changes. These data seemed less convincing and felt more tangential to the primary conclusions of PRMT2 influencing Tat function.

Responses to the Reviewers' Comments

We appreciate the valuable feedback provided by the reviewers regarding our manuscript. Their insightful suggestions have greatly contributed to the enhancement of the quality of our work. In response to their constructive advice, we have revised the manuscript by incorporating additional experimental studies, making necessary modifications to the text and figures, and expanding our discussions to provide deeper insights into our findings. Below, we present a comprehensive point-by-point response to address each of the reviewers' queries:

REVIEWER COMMENTS

Reviewer #1 (Remarks to the Author):

This work innovatively identified that PRMT2 plays a role in promoting HIV latency. It further elucidated a potentially new mechanism that PRMT2 interacts with HIV Tat protein and leads to its methylation at R52, which attenuates Tat incorporation into the nucleoplasmic AFF4 droplets, thus reducing Tat-SEC complex and preventing Tat-mediated HIV proviral transactivation. Overall, the results are convincing, which confirms PRMT2 as a novel host epigenetic factor that silences HIV proviruses. However, there are a few major issues to solve before the consideration for publication.

Response: We appreciate the reviewer for acknowledging the novelty of our work and for highlighting areas that require improvement in our manuscript. The constructive feedback has guided us in refining the quality and impact of our study during the revision process. In response to the reviewer's suggestions, we have undertaken additional experiments to address these concerns.

1). Most mechanistic studies were performed only in cell lines. It needs to show that PRMT2 indeed interacts with Tat protein and methylates Tat at R52 in primary CD4⁺ T cells. Likewise, the impact of PRMT2 on Tat-SEC/AFF4 protein complex also needs to be determined in primary CD4⁺ T cells.

Response: We appreciate the review's critical points. In order to investigate the potential interaction between Tat and PRMT2 and the methylation of R52 in primary CD4⁺T cells, we isolated CD4⁺T cells from the PBMCs of healthy donors. Subsequently, we infected these cells with a dual-color HIV-1_{GKO} reporter virus, wherein the Tat protein is expressed under the regulation of LTR. Utilizing our custom Tat antibody, co-immunoprecipitation experiments revealed a reciprocal interaction between Tat and PRMT2 in primary CD4⁺T cells. Furthermore, employing the specific TatR52^{me2} antibody, we established that the depletion of PRMT2 results in a considerable reduction in Tat methylation at R52. Importantly, this reduction occurs without any discernible impact on the overall abundance of Tat protein within primary CD4⁺T cells.

To assess the influence of PRMT2 on Tat's interaction with the SEC, we conducted an experiment involving the transduction of primary CD4⁺T cells with lentiviruses carrying control and PRMT2-targeting shRNAs. These transduced cells were

subsequently infected with HIV-1_{GKO} reporter viruses. Through immunoprecipitation analyses, we discovered that the depletion of PRMT2 significantly enhances the quantity of co-precipitated SEC subunits. This finding strongly indicates that PRMT2 exerts a negative regulatory effect on the Tat-SEC association within primary CD4⁺T cells.

We have included the outcomes of these investigations in the revised manuscript as new Figs. 4b, 4c, 4i, and 5k and have elaborated on them in the main text.

2). There was no result to show that CRISPR causes the complete knockout of endogenous PRMT2.

Response: We apologize for any confusion caused by the lack of clarity in our description of the CRISPR-mediated PRMT2 knockout results presented in Figs. 2f and 5f of the previous manuscript. The immunoblotting results were obtained using whole cellular extracts from pooled cells that were transduced with a pair of sgRNAs targeting distinct loci of the PRMT2 gene. It is expected and reasonable to observe that certain transduced cells still possess an intact PRMT2 gene and consequently produce the protein, which is detectable through immunoblotting.

Upon comparing the residual PRMT2 levels in E4 and 293T cells following transduction with the same pair of PRMT2 sgRNAs, as depicted in the previously mentioned Figs. 2f and 5f, it becomes evident that the efficiency of CRISPR-mediated PRMT2 knockout is notably higher in adherent 293T cells in comparison to suspended E4 cells. We have clarified in the updated manuscript's figure legends that the immunoblotting analyses were conducted on pooled cells.

We appreciate your understanding and have taken measures to enhance the clarity of the presentation.

3). Certain key protein immunoblotting data were quite subtle and not convincing. Protein bands need to be quantified. Statistic analysis shall be further performed from multiple independent protein immunoblotting experiments.

Response: We thank the reviewer for raising these significant critiques. Following the recommendations, we conducted quantitative analysis of crucial immunoblotting data from three independent experiments using ImageJ. We have now depicted these quantifications beneath the respective gel bands and/or as bar graphs in Figs. 4g, 4h, 4i, 4k, 5e, 5f, 5g, 5i, 5j, 5k, 6f, 7c, 7d, 7k, 7l, S4f, S4g, S5f, S5i-5k, S6b, S6k, S7a, S7b and S7f of the updated manuscript. These statistical insights further strengthen our elaboration on these results.

4). Earlier studies showed that JQ1 fails to reactivate latent HIV in patients' PBMCs. However, in Fig 3J, JQ1 was effective to reactivate latent HIV in primary CD4⁺ T cells isolated from HIV⁺ subjects. The authors shall discuss such discrepancy.

Response: We appreciate the reviewer for highlighting this inconsistency. JQ1, known for its high specificity and potency as an inhibitor of the BET bromodomain, has been

shown to consistently reactivate latent HIV-1 through counteracting BRD4's inhibitory effect on Tat transactivation, as demonstrated across various cell line models of HIV-1 latency (PMID: 23087374, 22802445).

However, the effectiveness of JQ1 in reactivating HIV-1 latency exhibits significant variability when applied to ex vivo CD4⁺ T cells isolated from individuals with HIV-1 infections receiving suppressive antiretroviral therapy (ART). Notably, substantial induction of HIV-1 transcripts is observed only in a limited subset of patients' CD4⁺ T cells following a 24-hour ex vivo treatment. This divergent reactivation of HIV-1 latency within patient cells can likely be attributed to the complex and heterogeneous nature of the latent reservoir of HIV-1 within the CD4⁺ T cells. Recent studies involving single-cell epigenetic and cell surface protein profiling have unveiled the profound cellular heterogeneity within this reservoir and demonstrated substantially variable response to the proviral reactivation by different families of LRAs (PMID: 36536105, 31425551).

Within these heterogeneous pools of latent reservoirs, a multitude of mechanisms may coexist and interact to maintain the latency of HIV-1 proviruses within memory CD4⁺ T cells. These mechanisms collectively contribute to the observed variation in the reactivation of latent HIV-1 virus within ex vivo CD4⁺ T cells obtained from different patients. Many previous investigations involved a 24-hour stimulation of PBMCs or CD4⁺ T cells using JQ1, revealing varying degrees of reactivation among distinct individuals with HIV-1 infections (PMID: 30645648, 22802445, 29714165, etc.).

In contrast, our study extended the treatment duration to 72 hours, focusing on CD4⁺ T cells from HIV-1 infected individuals. This extended treatment period likely contributes to the consistent and effective reactivation of latent HIV-1 within CD4⁺ T cells obtained from four patients. A thorough analysis of this disparity has been incorporated into the revised manuscript (From line 677 to 693).

5). It seems that no PRMT2-specific inhibitors are currently available. Thus, its therapeutic potential for HIV lytic reactivation and “shock and kill” cure is not clear. The authors shall discuss this issue.

Response: The reviewer's assessment is accurate. In relation to selective inhibitors for PRMT family members, only small-molecule compounds targeting PRMT1, -3, -4, -5, and -6 have been formulated and assessed in either in vitro cell lines or mouse models of cancer (PMID: 26789238 and 29781349). Notably, a specific inhibitor for PRMT2 is currently unavailable. However, fortunate are we that prior research has elucidated the atomic structure of PRMT2 in zebrafish and mice. Leveraging this insight, we are collaborating with colleagues who specialize in drug development. Together, we aim to craft agents with the capability to selectively inhibit PRMT2, thereby potentially reversing HIV-1 latency. This matter has been addressed in the Discussion section of the revised manuscript (From line 695 to 706).

Reviewer #2 (Remarks to the Author):

The paper by Jin et al focuses on the role of PRMT2, a methyltransferase, and its potential to inhibit Tat mediated HIV transcription which impacts establishment and maintenance of latency. By knocking down and overexpressing PRMT2 in variety of cell lines and primary cells they provide biochemical and functional evidence that PRMT2 influences Tat function. They biochemically map key domains and key interacting factors which include components of the Super Elongation Complex. Finally, they make an argument that a role of PRMT2 is to influence nuclear distribution of these factors. Strengths of the paper are the exhaustive mapping of biochemical activities and interactions of PRMT2 with Tat and the SEC, the use of Tat and PRMT2 mutations to validate knock-down and overexpression experiments, and the use of primary cells, including clinical samples that support the argument for PRMT2 influencing latency. Modest weaknesses include that there are limited data, examining the role of endogenous PRMT on HIV infected cells, there may be a missed opportunity to explore potential functional interactions between PRMT2 and PRMT6, a previous methyltransferase described to regulate HIV transcription, and, although nuclear distribution of Tat and AFF4 were convincing, the PRMT2 role was less so, making this feel tangential to the overall PRMT2 story.

Response: We are sincerely grateful to the reviewer's positive feedback and insightful critiques, which have played a pivotal role in fortifying our arguments and enhancing the overall quality and impact of our manuscript.

Some specific comments are below.

1. All experiments were performed with cells that had factors overexpressed or knocked down. Although these experiments are essential for getting to mechanism and are convincing, it would validate these results if experiments were performed demonstrating Tat methylation, co-IPs, ChIPs, etc, using infected cells that expressed normal endogenous levels of PRMT2.

Response: We appreciate the reviewer for providing such constructive comments. However, due to the limited availability of CD4⁺ T cells obtained from HIV-1 infected patients under antiretroviral therapy (ART), it becomes impractical for us to delve into the intricate mechanisms underlying PRMT2's role in controlling HIV-1 latency using patient-derived cells.

In order to corroborate our mechanistic discoveries within a cellular context that closely mimics patient CD4⁺ T cells, we examined the interaction between Tat and PRMT2, the methylation status of Tat, as well as the integrity of the Tat-SEC complex in primary CD4⁺T cells that were obtained from healthy donors and transduced with an dual-fluorescence HIV-1 reporter virus. The outcomes obtained from these cells recapitulate our earlier findings from models of HIV-1 latency established in immortalized cell lines. We presented these new results as new figures (4b, 4c, 4i, and

5k) in the revised manuscript.

2. Expression of PRMT6 should be examined as a control to assure that PRMT2 is not influencing the expression of this PRMT family member. Related to this, examining the physical and functional interactions of PRMT2 and PRMT6 in HIV infected cells seems like an important question that could have been addressed.

Response: This is indeed a crucial aspect that we inadvertently overlooked in the previous manuscript. To investigate the potential impact of PRMT2 on PRMT6 expression, we scrutinized the immunoblotting data, presented as Fig. 5f in our previous manuscript. This investigation led to the revelation that the depletion of PRMT2 does not elicit any discernible effects on the expression of PRMT6 in 293T cells. Furthermore, PRMT2 depletion do not substantially alters PRMT6 level in primary CD4⁺T cells that were derived from healthy donors and subsequently transduced with dual-fluorescence HIV-1_{GKO} reporter virus (Fig. 5k). Thus, we can conclude that PRMT2's influence does not extend to altering the levels of PRMT6. The observed effects of PRMT2 depletion on Tat methylation and the integrity of the Tat-SEC complex are direct and independent of any alternations in PRMT6.

To investigate the potential physical interaction between PRMT2 and PRMT6, we conducted a series of experiments involving the expression of PRMT2 and PRMT6 individually, as well as in combination with Tat, within 293T cells. Subsequently, an immunoprecipitation assay was performed to elucidate any binding interactions. These analyses revealed that when exogenous PRMT2 was expressed independently of Tat co-expression, there was no detectable co-precipitation with endogenous PRMT6. However, in the presence of Tat, a minor fraction of endogenous PRMT6 and PRMT2 was co-precipitated with exogenous PRMT2 and PRMT6, respectively. This observation underscores that while PRMT2 and PRMT6 do not exhibit a direct affinity for each other, they can indeed interact indirectly through their shared substrate, Tat protein (Fig. R1).

Fig. R1. 293T cells were transfected with flag-tagged PRMT2, PRMT6 alone or in combination with myc-tagged Tat followed by immunoprecipitation with anti-flag agarose and immunoblotting with indicated antibodies.

3. In fig 2 the reversal of latency takes several days. This seems somewhat surprising in the context of transcriptional regulation. Is there a selection of cells or some change in cell phenotypes observed with the diminished PRMT2 expression? Does this provide any insights into PRMT2 function? Also, figure 2 I and J were unclear as to what was 100% since they are observing maybe only 20% infected cells; how were these profiles generated?

Response: We appreciate the reviewer for raising these critical issues. In terms of the temporal aspect related to the reversal of HIV-1 latency, it is important to acknowledge that the process of reactivating latent HIV-1 virus following the depletion of specific host restriction factors, such as LEDGF/p75 (PMID: 25590759, 31616795, 32426500), FOXO1 (PMID: 32541947), EZH2 (PMID: 18829756), among others, in cell line models of HIV-1 latency, typically spans several days. This contrasts significantly with the swift reactivation of proviral DNA observed upon T cell activation. Thus, we assumed that the extent of decrease in PRMT2 levels reaches the threshold to render both molecular and cellular changes that are visible in our assays.

There exist several plausible mechanisms that contribute to the slow kinetics of latency reversal elicited by the sole depletion of a single host restriction factor. Foremost, the intricate landscape of HIV-1 latency entails diverse molecular mechanisms operating at multiple levels to silence proviral gene expression, thereby effectively suppressing latent HIV-1 reactivation. Even upon the removal of a singular barrier through the depletion of a specific host restriction factor, several residual barriers persist, contributing to the gradual reactivation kinetics of latent HIV-1 in host cells. Secondly, prior reports indicated that knockdown of distinct host restriction factors can indeed reactivate latent HIV-1 provirus within different latently infected cell lines; however, the magnitude and tempo of this induction exhibit substantial variations (PMID: 21715480). This disparity suggests a differential contribution of individual factors in both establishing and maintaining HIV-1 latency, subsequently resulting in varying kinetics of latency reversal upon their depletion. Lastly, while we ensured the analysis of virally transduced cells through a two-day puromycin selection, it is crucial to acknowledge that the degree of PRMT2 depletion may not be uniform. This could result in only a subset of infected cells exhibiting a reduction in PRMT2 levels to the requisite threshold for discernible reactivation of latent HIV-1. The heterogeneous reduction of PRMT2 levels following knockdown could therefore account for the inefficiency and gradual nature of HIV-1 provirus reactivation.

In essence, these multifaceted factors contribute collectively to the observed variability and gradual reactivation kinetics of latent HIV-1 upon depletion of specific host restriction factors, shedding light on the intricate nature of HIV-1 latency regulation.

In order to assess the impacts of PRMT2 depletion on latently infected HIV-1 cell lines, we evaluated cell growth and viability at various time intervals subsequent to PRMT2 knockdown. Our analyses demonstrated that the depletion of PRMT2 does

not yield any discernible impacts on the proliferation and survival of these cells (Fig. R2). As a result, we concluded that the reactivation of latent HIV-1 following PRMT2 depletion is not attributed to the phenotypic alterations within the host cells. This underscores the notion that PRMT2 plays a direct role in influencing the latency reactivation process, devoid of the consequential effects on the general cellular growth and viability.

Fig. R2 Growth and viability of HIV-1 cells post PRMT2 depletion. **a** Growth curve of E4 cells at different time points post lentiviral transduction of a control or PRMT2 shRNA. **b** Flow cytometry pseudocolor plots showing the population of 2D10 cells with reactivated HIV-1 provirus at different time points post PRMT2 depletion.

In Figs. 2i and 2j, we delved into the potential involvement of PRMT2 in the re-establishment of HIV-1 latency subsequent to reactivation. To this goal, we successfully achieved a remarkable reactivation of latent HIV-1 virus, approximately reaching 98%, by subjecting both control and PRMT-depleted 2D10 cells to an overnight treatment with 10 ng/ml TNF α (at D0). Following this reactivation, we monitored the population of reactivated cells at distinct time points (D3, D6, and D9) subsequent to the TNF α withdrawal.

For the purpose of our quantification, we employed a reference point of 100%, representing the scenario where all 2D10 cells harbored active HIV-1 provirus after the overnight TNF α treatment. This approach allowed us to precisely gauge the extent of reactivation and the subsequent kinetics of latency re-establishment in both control and PRMT2-depleted cells.

4. Experiments show changes in Tat-mediated HIV-1 transcription but Tat still induces some transcription. Were experiments performed to provide insights into the impact PRMT2 has on HIV-1 production.

Response: Yes, the reviewer is right. From the experimental data at hand, our intended claim is that the methylation of Tat at R52 by PRMT2 serves as a preventive mechanism against proviral reactivation and subsequent HIV-1 production. This effect

is achieved through two interconnected mechanisms:

Nucleolar Sequestration: PRMT2-mediated methylation of Tat at R52 contributes to the sequestration of Tat within the nucleolus, leading to the restriction of its involvement in transcriptional activity.

Attenuated Incorporation into Nucleoplasmic SEC Droplets: Furthermore, this methylation event hinders the efficient incorporation of Tat into nucleoplasmic SEC droplets. Consequently, this limitation curtails Tat's transcriptional activity, thereby playing a pivotal role in suppressing proviral reactivation.

The combination of the two interconnected mechanisms underscores the intricate regulatory role of PRMT2-mediated Tat methylation in orchestrating the repression of HIV-1 transcriptional activity and subsequent viral production.

5. The hypothesis that PRMT2 influences Tat and AFF4 phase separation are intriguing. However, the blots in Fig 7 for PRMT2 suggested subtle changes. These data seemed less convincing and felt more tangential to the primary conclusions of PRMT2 influencing Tat function.

Response: We appreciate both the positive and critical comments provided by the reviewer. In assessing the extent of changes in PRMT2 levels as depicted in Fig. 7d, we determined the relative expression levels of PRMT2 by conducting densitometry analysis of the gel bands. These values were then normalized against the Lamin A/C signal and are presented beneath each respective band.

While the PRMT2 level remains detectable subsequent to PRMT2 knockdown, a noticeable reduction in its magnitude is evident. Particularly significant is the extent of this reduction in PRMT2, which has proven adequate in inducing proviral reactivation across diverse cell lines, primary CD4⁺ T cell models of HIV-1 latency, as well as in HIV-1 infected CD4⁺ T cells obtained from patients under ART treatment. The primary conclusion of this work is that PRMT2 promotes Tat nucleolar sequestration and thus prevent its incorporation into nucleoplasmic SEC droplets to suppress HIV-1 transcription and proviral reactivation in latently infected cells. The influence of PRMT2 on Tat and AFF4 phase separation offers a mechanistic insight into PRMT2's action in the establishment and maintenance of HIV-1 latency.

Reviewers' Comments:

Reviewer #1:

Remarks to the Author:

The authors have adequately addresses my questions.

Reviewer #2:

Remarks to the Author:

I find the authors addressed the major concerns of the original manuscript.

Responses to the Reviewers' Comments

We appreciate both reviewers for evaluating our manuscript again. Both of them think that the revised manuscript have addressed their concerns raised during the first-round of evaluation.

REVIEWERS' COMMENTS

Reviewer #1 (Remarks to the Author):

The authors have adequately addresses my questions.

Response: Thanks the reviewer for taking time to review our revised manuscript.

Reviewer #2 (Remarks to the Author):

I find the authors addressed the major concerns of the original manuscript.

Response: Thanks the reviewer for taking time to review our revised manuscript.